# LANGUAGE MODELS USE LOOKBACKS TO TRACK BELIEFS

**Nikhil Prakash$^{\diamond}$, Natalie Shapira$^{\diamond}$, Arnab Sen Sharma$^{\diamond}$, Christoph Riedl$^{\diamond}$,
Yonatan Belinkov$^{\spadesuit}$, Tamar Rott Shaham$^{\heartsuit}$, David Bau$^{\diamond}$, Atticus Geiger$^{\clubsuit\dagger}$**

$^{\diamond}$Northeastern University    $^{\spadesuit}$Technion    $^{\heartsuit}$MIT CSAIL    $^{\clubsuit}$Goodfire    $^{\dagger}$Pr(Ai)$^2$R Group

## ABSTRACT

How do language models (LMs) represent characters' beliefs, especially when those beliefs may differ from reality? This question lies at the heart of understanding the Theory of Mind (ToM) capabilities of LMs. We analyze LMs' ability to reason about characters' beliefs using causal mediation and abstraction. We construct a dataset, *CausalToM*, consisting of simple stories where two characters independently change the state of two objects, potentially unaware of each other's actions. Our investigation uncovers a pervasive algorithmic pattern that we call a *lookback mechanism*, which enables the LM to recall important information when it becomes necessary. The LM binds each character-object-state triple together by co-locating their reference information, represented as Ordering IDs (OIs), in low-rank subspaces of the state token's residual stream. When asked about a character's beliefs regarding the state of an object, the *binding lookback* retrieves the correct state OI and then the *answer lookback* retrieves the corresponding state token. When we introduce text specifying that one character is (not) visible to the other, we find that the LM first generates a *visibility ID* encoding the relation between the observing and the observed character OIs. In a *visibility lookback*, this ID is used to retrieve information about the observed character and update the observing character's beliefs. Our work provides insights into belief tracking mechanisms, taking a step toward reverse-engineering ToM reasoning in LMs.

## 1 INTRODUCTION

Theory of Mind (ToM), the ability to infer others' mental states, is an essential aspect of social and collective intelligence (Premack & Woodruff, 1978; Riedl et al., 2021). Recent studies have established that LMs can solve some tasks requiring ToM reasoning (Street et al., 2024; Strachan et al., 2024a; Kosinski, 2024), while others have highlighted shortcomings (Ullman, 2023; Sclar et al., 2025; Shapira et al., 2024, *inter alia*). Previous studies primarily rely on behavioral evaluations, which do not shed light on the internal mechanisms by which LMs encode and manipulate representations of mental states to solve (or fail to solve) such tasks (Hu et al., 2025; Gweon et al., 2023).

In this work, we examine *how LMs internally represent and track beliefs* of characters, a core aspect of ToM (Dennett, 1981; Wimmer & Perner, 1983). A classic example is the Sally-Anne test (Baron-Cohen et al., 1985), which evaluates ToM in humans by assessing whether individuals can track conflicting beliefs: Sally's belief, which diverges from reality because of missing information, and Anne's belief, which is updated based on new observations. Our goal is to determine whether LMs learn a systematic solution to such tasks or rely on superficial statistical association.

We construct *CausalToM*, a dataset of simple stories involving two characters, each interacting with an object to change its state, with the possibility of observing one another. We then analyze the internal mechanisms that enable `Llama-3-70B-Instruct`, `Llama-3.1-405B-Instruct` and `Qwen2.5-14B-Instruct` (Grattafiori et al., 2024; Bai et al., 2023) to reason about and answer questions regarding the characters' beliefs about the state of each object (for a sample story, see Section 3 and for the full prompt refer to Appendix A).

---

Correspondence to `prakash.nik@northeastern.edu`.

Our findings provide strong evidence for a systematic solution to belief tracking. We discover that LMs use a pervasive computation, which we refer to as the *lookback mechanism*, for belief tracking. This mechanism enables the model to recall important information at a later stage. In a lookback, two copies of a single piece of information are transferred to two distinct tokens. This allows attention heads at the latter token to look back at the earlier one when needed and retrieve vital information stored there, rather than transferring it directly (see Fig. 1).

We identify three key lookback mechanisms that collectively perform belief tracking: 1) *Binding lookback* (Fig. 3(i)): First, the LM assigns *ordering IDs* (OIs; Dai et al. 2024) that encode whether a character, object, or state token appears first or second. Then, the character and object OIs are copied to the corresponding state token and the final token residual stream. Later, when the LM needs to answer a question about a character's beliefs, it uses this information to retrieve the answer state OI. 2) *Answer lookback* (Fig. 3(ii)): Uses the answer state OI from the binding lookback to retrieve the answer state token value. 3) *Visibility lookback* (Fig. 7): When a visibility condition between characters is mentioned, the model employs additional reference information called the *visibility ID* to retrieve information about the observed character, augmenting the observing character's awareness.

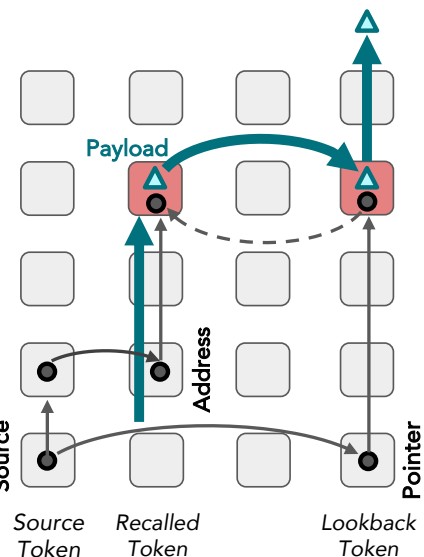

Figure 1: **The lookback mechanism** performs conditional reasoning; The *source token* contains reference information that is copied into two instances, creating a *pointer* and an *address*. Next to the address in the residual stream is a *payload*. When necessary, the model retrieves the payload by dereferencing the pointer. Solid lines represent information flow, while the dotted line indicates the attention "looking back" from pointer to address.

Overall, this work not only advances our understanding of the internal computations in LMs that enable ToM but also uncovers a pervasive mechanism that plays a foundational role for in-context reasoning. All code and data supporting this study are available at `https://belief.baulab.info`.

## 2 THE LOOKBACK MECHANISM

Our investigation uncovers a recurring pattern of computation that we call the *lookback mechanism*.[1] In lookback, a *source reference* is copied (via attention) into an *address* copy in the residual stream of the *recalled token* and a *pointer* copy in the residual stream of the *lookback token* that occurs later in the text. The LM places the address alongside a *payload* in the recalled token's residual stream that can be brought forward to the lookback token if necessary. Fig. 1 shows a generic lookback.

That is, the LM can use attention to dereference the pointer and retrieve the payload present in the residual stream of the recalled token (which might contain aggregated information from previous tokens), bringing it to the residual stream of the lookback token. Specifically, the pointer at the lookback token forms an attention query vector, while the address at the recalled token forms a key vector. The pointer and address are not necessarily exact copies of the source reference, but they do have a high dot product after being transformed by a query or key attention matrix, respectively. Hence, a *QK-circuit* (Elhage et al., 2021) is established, forming a bridge from the lookback token to the recalled token. The LM uses this bridge to move the payload that contains information needed to

---

[1]Although this mechanism may resemble *induction heads* (Elhage et al., 2021; Olsson et al., 2022), it differs fundamentally. In induction heads, information from a previous token occurrence is passed only to the subsequent token, without being duplicated to its next occurrence. In contrast, the lookback mechanism copies the same information not only to the location where the vital information resides but also to the target location that needs to retrieve that information.

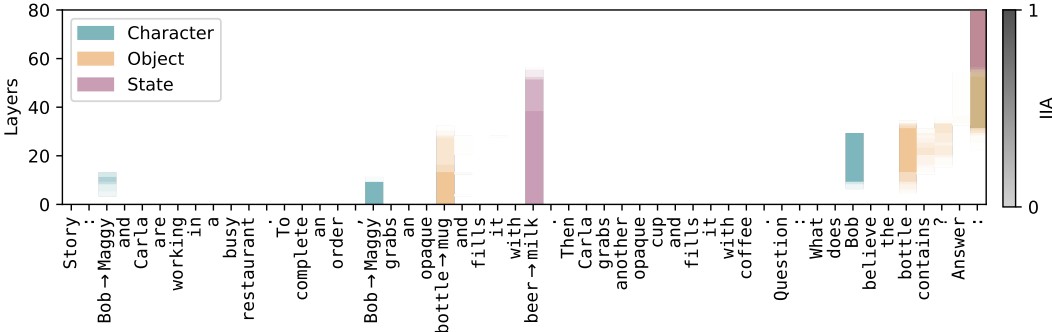

Figure 2: **Tracing information flow** of crucial input tokens using causal mediation analysis.

complete the subtask through the *OV-circuit*. See Appendix C for background on the residual-stream framework as well as on QK- and OV-circuits.

To develop an intuition for why an LM would learn to implement lookback mechanisms, consider that during training, LMs process text in sequence with no foreknowledge of what might come next. Instead of trying to resolve every possible future question about the current context, it would be useful to place addresses alongside payloads that might be useful to remember in the future when performing a variety of downstream tasks. In our setting, the LM constructs a representation of a story without any certainty about the questions it may later be asked about that story, so the LM localizes pivotal information in the residual stream of certain tokens, which later become payloads and addresses. When the question text is reached, pointers are constructed that reference this crucial story information and dereference it to find an answer to the question.

## 3 EXPERIMENTAL SETUP: DATASET, MODELS, AND METHODS

### 3.1 DATASET

Existing datasets for evaluating ToM capabilities of LMs are designed for behavioral testing and lack counterfactual pairs needed for causal analysis (Kim & Sundar, 2012). To address this problem, we construct *CausalToM*, a structured dataset of simple stories, where each story involves two characters, each interacting with a distinct object causing the object to take a unique state. For example: "`Character1` and `Character2` are working in a busy restaurant. To complete an order, `Character1` grabs an opaque `Object1` and fills it with `State1`. Then `Character2` grabs another opaque `Object2` and fills it with `State2`." We then ask the LM to reason about one of the characters' beliefs regarding the state of an object: "`What does` `Character1` believe `Object2` contains?" We analyze the LM's ability to track characters' beliefs in two distinct settings. (1) *No Visibility*, where both characters are unaware of each other's actions, and (2) *Explicit Visibility*, where explicit information about whether a character can/cannot observe the other's actions is provided, e.g., "`Bob` can observe `Carla`'s actions. `Carla` cannot observe `Bob`'s actions." We also provide general task instructions (e.g., answer `unknown` when a character is unaware); refer to Appendices A & B for the full prompt and additional dataset details. All subsequent experiments are conducted on 80 samples that the model answers correctly. We also demonstrate generalization of the mechanism to BigToM dataset (Gandhi et al., 2024) in Appendix M.

### 3.2 MODELS

Our experiments analyze `Qwen2.5-14B-Instruct`, `Llama-3-70B-Instruct`, and `Llama-3.1-405B-Instruct` models in FP32, FP16, and INT8 precision, respectively, using *NNsight* (Fiotto-Kaufman et al., 2025). Results for `Qwen2.5-14B-Instruct` and `Llama-3.1-405B-Instruct` can be found in Appendix N. We selected these models due to their strong behavioral performance under both no-visibility and explicit-visibility conditions. Additional details can be found in Appendix D.

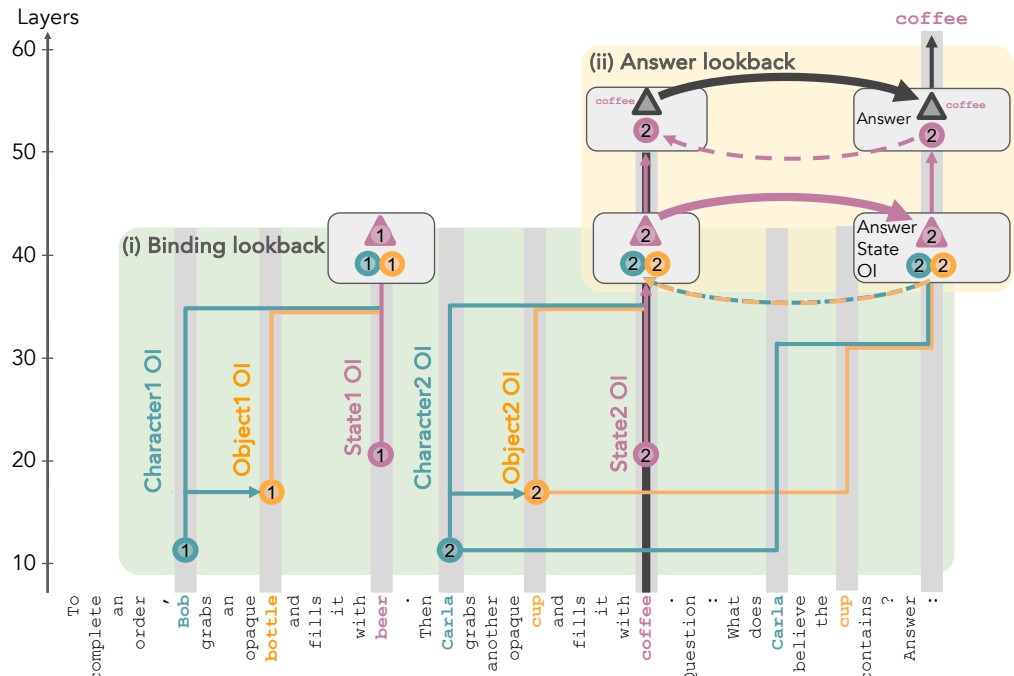

Figure 3: **Belief Tracking with *no visibility* between characters.** We hypothesize that the LM tracks beliefs using two lookback mechanisms. First, in (i) **Binding lookback**, LM binds together each character-object-state triple in the state token residual stream. When asked about a specific character-object pair, the LM looks back to the corresponding OIs to retrieve the correct state OI. Second, in (ii) **Answer lookback**, LM dereferences that state OI (used as a pointer) to retrieve the token value of the correct state. Colors indicate information type, shapes indicate role of information in lookback (see Fig. 1), e.g., state OI is a payload (▲) in (i) and a pointer-address (⬤) in (ii).

## 3.3 CAUSAL MEDIATION ANALYSIS

Our goal is to develop a mechanistic understanding of how LMs reason about characters' beliefs and answer related questions (Saphra & Wiegreffe, 2024). A key method for conducting causal analysis is *interchange interventions* (Vig et al., 2020; Geiger et al., 2020; Finlayson et al., 2021), in which the LM is run on paired examples: an *original input* **o** and a *counterfactual input* **c**, and certain internal activations in the LM run on the original input are replaced with those computed from the counterfactual, a process also known as activation patching. We begin our analysis by tracing information flow from key input tokens to the final output, by performing interchange interventions on the residual vectors. Specifically, we construct an intervention dataset where **o** contains a question about the belief of a character not mentioned in the story, while the story in **c** includes the same queried character, as shown in Fig. 2. The expected outcome of this intervention is a change in the final output of **o** from *unknown* to a state token, such as beer. We conduct similar interchange interventions for object and state tokens (refer to Appendix E for details).

Figure 2 presents the aggregated results of this experiment for the key input tokens Character1, Object1, and State1. The cells are color-coded to indicate the *interchange intervention accuracy* (IIA; Geiger et al., 2022). Even at this coarse level of Causal Mediation Analysis (Mueller et al., 2024; Vig et al., 2020; Meng et al., 2022), several significant insights emerge: 1) Information from the correct state token (beer) flows directly from its residual stream to that of the final token in later layers, consistent with prior findings (Lieberum et al., 2023; Prakash et al., 2024); 2) Information associated with the query character and the query object is retrieved from their earlier occurrences and passed to the final token before being replaced by the correct state token.

**Desiderata Based Patching via Causal Abstraction**   The causal mediation experiments provide a coarse-grained analysis of where information flows, but do not identify what information is being transferred. In a transformer, the first layer represents the input and the last layer represents the output, but we wish to know: what is represented in the middle? We analyze the internal mechanism

using *Causal Abstraction* (Geiger et al., 2021; 2024); First, we hypothesize a high-level causal model of the computational steps from input to output (Sec. 4), and then align its variables with the LM's internal activations (Sec. 5). We test the alignment through targeted interchange interventions on causal variables in the hypothesized model and hidden activations in the LM. If the LM produces the same output as the causal model under these aligned interventions, it provides evidence supporting the hypothesized causal model. We quantify this effect using *interchange intervention accuracy* (IIA; Geiger et al., 2022), which measures the proportion of cases where the intervened causal model and intervened LM agree. See Appendix F for more details.

In addition to measuring IIA on entire residual stream vectors, we also intervene on localized subspaces to further isolate causal variables. To identify the subspace of a specific variable, we employ *Desiderata-based Component Masking* (De Cao et al., 2020; Davies et al., 2023; Prakash et al., 2024). This method learns a sparse binary mask over the activation space that maximizes the logit of the hypothesized causal model output. We train a mask to select singular vectors of the activation space that encode a high-level variable (see Appendix H for details). Our experiments in Sec. 5 report both interventions on the full residual stream and on the identified subspaces.

## 4 HYPOTHESIZED HIGH-LEVEL CAUSAL MODEL OF BELIEF TRACKING

Here we start with an overview of our hypothesized causal model of belief-tracking when characters are not aware of each other's actions. The causal model is an algorithmic process that has variables with structural roles that do not refer to the details of a transformer architecture. Appendix G presents the full pseudocode of the causal model. In Section 5, we will present experiments to verify that the causal model's variables align with representations in the transformer.

Belief tracking begins when the causal model assigns *ordering IDs* (OIs; ◉, ◉, ◉) to each character, object, and state token, marking their order of appearance. For instance, in the example in Fig. 3, `Bob` is assigned first character OI (①), and `Carla` is assigned the second character OI(②). Please refer to Appendix C for more information about OI. Then it uses these OIs in two lookback mechanisms:

**(i) Binding lookback.** The causal model creates address copies of each character OI (◉) and object OI (◉) that are bound to the state OI (Binding Payload, ▲), creating a character–object–state triple. When a question is asked about a character and object, the causal model creates pointer copies of that character and object OIs (Binding Pointers ◉, ◉) and dereferences them to retrieve the state OI.

**(ii) Answer lookback.** The causal model creates an address copy of the state OI (Answer Address ◉) that is bound to the state token (Answer Payload, ▲). Through the binding lookback, a pointer copy of this OI (◉) is created. The causal model dereferences the pointer to retrieves the correct state token payload as the final output.

## 5 VERIFYING THE HYPOTHESIZED CAUSAL MODEL OF BELIEF TRACKING

We test our hypothesized causal model by localizing its variables within the transformer's neural representations. Specifically, we localize the addresses, pointers, and payloads of the (i) binding lookback and (ii) answer lookbacks within the LM's internal activations. In Fig. 3, we show a trace of the causal model run on an input overlayed onto a schematic of a transformer architecture. This visualizes the alignment between variables in the causal model and locations in the LM residual stream that the experiments in the remaining of this paper will support. In the binding lookback, the character and object OI addresses are realized in the residual stream of the state token. The pointer copies are brought forward to the last token residual stream where they are dereferenced via attention to bring forward the correct state OI payload. In the answer lookback, the address copy of the state OI is in the state token residual stream while the pointer copy is in the last token residual stream.

Each of the following experiments localizes the presence of specific ordering IDs (OIs) and verifies their roles as hypothesized by our causal model. We do this by targeted interchange intervention experiments on the causal model and the LM. We copy hidden states between identical tokens (for example, replacing the representation of ":" in one context with the representation of ":" in another context, as in Fig. 4). When this intervention causes the LM's have the same output as the causal model under an interchange intervention on OI variables, we have evidence that the OI is carrying out

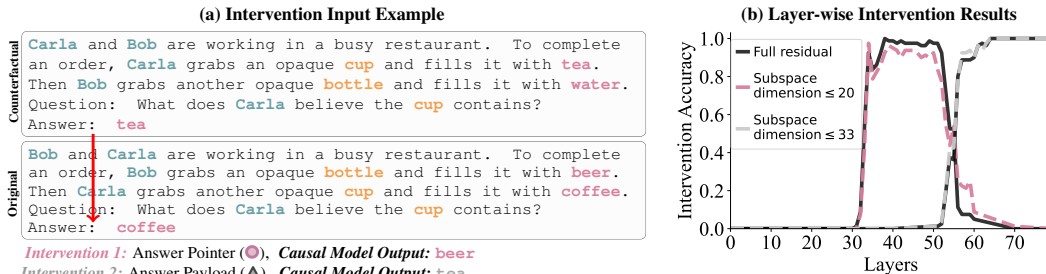

Figure 4: **Answer Lookback Pointer and Payload**: The causal model predicts that if we alter the "Answer Payload △" of the original to instead take the value of the counterfactual answer payload, the output should change from **coffee** to **tea**; the gray curve in the line plot shows this does occur when patching residual vectors at the ":" token beyond layer 56, providing evidence that the answer payload resides in those states. On the other hand the causal model predicts that taking the counterfactual "Answer Pointer ⬤" would change the original run output from **coffee** to **beer**—a new output that matches *neither* the original nor the counterfactual!—and we do see this surprising effect, again when patching layers between 34 and 52, providing strong evidence that the answer pointer is encoded at those layers. These results suggest the Answer Lookback occurs between layers 52 and 56.

the hypothesized role. Each experiment reports the effects of $n = 80$ different cases with the same structure, and the effect is measured at every layer.

Because the last step of the causal model is easiest to understand, we proceed through the experiments in reverse order, beginning with an experiment to verify the final "answer lookback" stage. After this instructive starting point, we work backward to verify the earlier steps of the model. Additional results can be found in Appendix I and J.

## 5.1 STEP II: ANSWER LOOKBACK – RETRIEVING THE CORRECT STATE

**Localizing the Answer Payload** We first verify the presence of the correct Answer Payload △ at the deepest layer representation of final token ":". To do so, we run an interchange intervention experiment shown in Fig. 4a in which the counterfactual example **c** swaps the order of the characters and objects of the original example **o** and also replaces the state (drinks) tokens with new values. If the Answer Payload is correctly localized, swapping it should cause the answer of the counterfactual (e.g., **tea**) to replace the answer of the original example (e.g., **coffee**). The gray line in Fig. 4b shows that this output change is observed in every one of $n = 80$ cases, both when intervening on the full residual stream and on the identified subspace. However, not at every layer: the information is only present after layer 56, indicating that before this stage, the transformer has not yet retrieved the correct answer payload into the residual stream. That is consistent with our hypothesis that at early steps, the OI has not yet been dereferenced. At an earlier stage, we expect to see an Answer Pointer.

**Localizing the Pointer Information** To identify the Answer Pointer ⬤ before it is dereferenced to bring the payload (state token value), we examine the representations of ":" at layers earlier than 56. Our causal model provides the hypothesis: if the Answer Pointer is present, then patching the pointer from the counterfactual run into the original run should redirect the LM to attend to the location of the correct counterfactual state and fetch its payload. For example, in Fig. 4a the counterfactual pointer references the first presented state. When we patch it into the original story, we expect the model's answer to change to **beer** rather than **coffee**. The colored line in Fig. 4b confirms that this effect is consistently observed when patching any layer between $34 - 52$ (both when patching the full residual stream and the identified subspace), supporting our hypothesis that these layers encode the Answer Pointer information at the final token, rather than directly transferring token values.

## 5.2 STEP I: BINDING LOOKBACK – LINKING CHARACTERS, OBJECTS, AND STATES

**Localizing the Address and Payload** In this experiment, we verify the presence of the address copies of the character and object OIs as well as the payload (state OI) at the state token residual stream (recalled token, Fig. 3). As illustrated in Fig. 5a, we construct an intervention dataset where

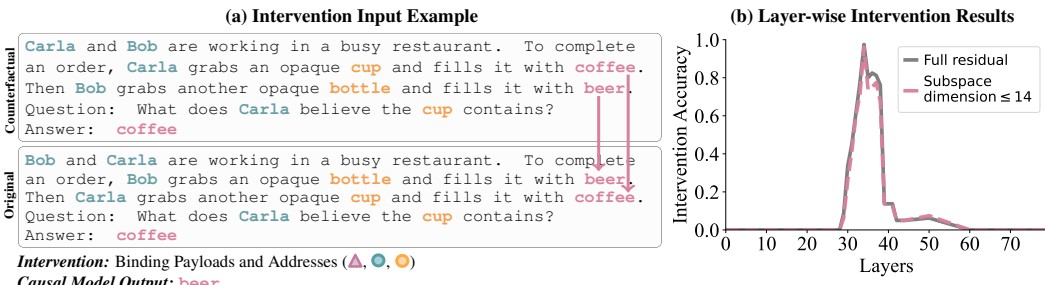

Figure 5: **Binding lookback Address and Payload:** The causal model predicts that swapping addresses (character and object OIs; ⬤ and ⬤) and payloads (state OIs; ▲) should cause the binding lookback mechanism to attend to the alternate state token and retrieve its state OI. This retrieved state OI is then dereferenced by the answer lookback, producing the corresponding token as the output (e.g., **beer** instead of **coffee**). The LM's behavior matches this prediction when we perform interchange interventions on the state token across layers 33–38. These findings support our hypothesis that both address and payload information are encoded in the residual stream of state tokens.

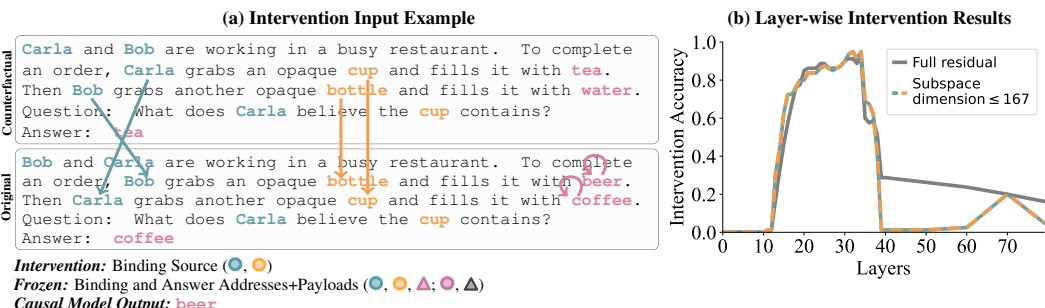

Figure 6: **Source Reference Information of Binding lookback**: The causal model predicts that swapping the source reference information (character and object OIs; ⬤, ⬤), while freezing the addresses and payloads of the binding lookback, should cause the binding lookback mechanism to attend to the alternate state token and retrieve its state OI, which would generate alternate state token as the final output via the answer lookback (e.g., **beer** instead of **coffee**). The LM's behavior matches this prediction when we perform interchange interventions at the character and object tokens across layers 20-34. These results support our hypothesis that source reference information is encoded in the residual stream of character and object tokens.

each example consists of an original input **o** with an answer that is not *unknown* and a counterfactual input **c** where the character, object, and state token values are identical, except the ordering of the two story sentences is swapped while the question remains unchanged. The expected LM's output predicted by our hypothesized causal model is the other state token in the original example, e.g., **beer**. That is because patching the address and payload values at each state token, without changing the pointer, makes the LM dereference the other state token. As a result, the model's output should flip to the other state token in the original input.

We perform the interchange intervention experiment layer-by-layer, where we replace the residual stream vector (or the identified subspace) of the first state token in the original run with that of the second state token in the counterfactual run and vice versa for the other state token. It is important to note that if the intervention targets state token values instead of their OIs, it should not produce the expected output. (This happens in the earlier layers.) As shown in Fig. 5b, the strongest alignment occurs between layers 33 and 38, supporting our hypothesis that the state token's residual stream contains both the address (character and object OIs) and the payload information (state OI).

**Localizing the Source Reference Information** Next, we localize the source reference information, i.e., character and object OIs at their respective token residual stream. As illustrated in Fig. 6a, we conduct an intervention experiment with a dataset where the counterfactual example, **c**, swaps

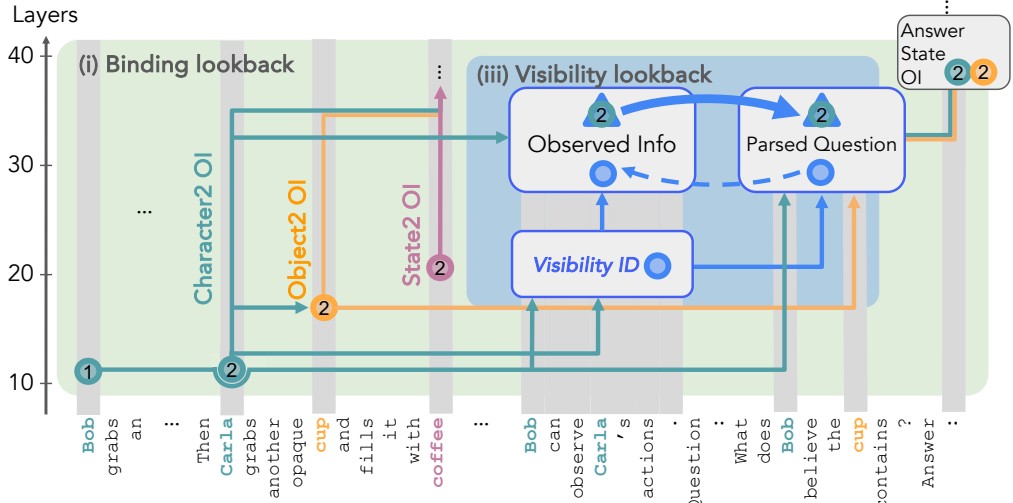

Figure 7: **Visibility Lookback** When one (observing) character can see another (observed) character, the LM assigns a visibility ID ( ) to the visibility sentence where this relation is defined. An address copy of this visibility ID remains in the visibility sentence's residual stream. A pointer copy of the visibility ID is transferred to the subsequent tokens' residual stream. The LM dereferences this pointer through a QK-circuit, bringing forward the payload ( ), when processing subsequent tokens. Based on initial evidence, this payload contains the observed character's OI( ). See Appendix K for details. This mechanism allows the model to incorporate the observed character's knowledge into the observing character's belief state, enabling more complex belief reasoning.

the order of the characters and objects as well as replaces the state tokens with entirely new ones while keeping the question the same as in **o**. Under this setup, an interchange intervention on the hypothesized causal model that targets the source reference should propagate changes through both the address and the pointer, leaving the final output unchanged. However, if we instead freeze the state token residual stream, which carries both the payload and the address, the causal model produces the alternate state token (e.g., beer in Fig. 6), as the pointer refers to the other state's address.

In the LM, we interchange the residual streams of the character and object tokens layer-by-layer, while keeping the residual stream of the state token fixed. As shown in Fig. 6b, this experiment reveals alignment between layers 20 and 34, indicating that source reference is encoded in the residual streams of the character and object tokens within this layer range. Additional results are provided in Appendix I, where Fig. 13 shows that freezing the residual stream of the state token is necessary for this alignment to emerge. These findings support our hypothesis that source reference is present in the character and object tokens and is subsequently transferred to the recalled and lookback tokens.

**Localizing the Pointer Information** Finally, we localize the pointer copies of the character and object OIs to their corresponding tokens in the question and to the final token. See Appendices I & J for details of the experiments and results.

In summary, belief tracking begins in layers 20–34, where character and object OIs are encoded in their respective token representations. These OIs are transferred to the corresponding state tokens in layers 33–38. When a question is asked, pointer copies of the relevant character and object OIs are moved to the final token by layer 34, where they are dereferenced to retrieve the correct state OI. At the final token, this state OI is represented across layers 34–52, and between layers 52–56, it is dereferenced to fetch the answer from the correct state token, producing the final output.

## 6 Impact of Visibility Conditions on Belief Tracking Mechanism

So far, we have demonstrated how the LM uses ordering IDs and two lookback mechanisms to track the beliefs of characters that cannot observe each other. Now, we explore how the LM updates the beliefs of characters when one character (*observing*) can observe the actions of the other (*observed*).

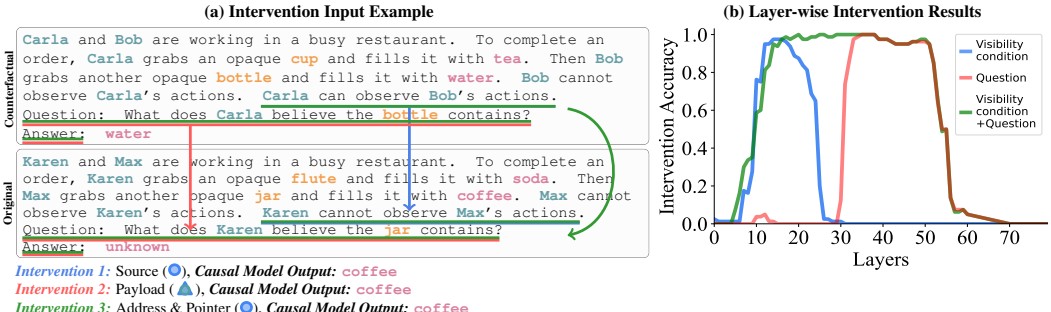

Figure 8: **Visibility Lookback**: We conduct three interchange intervention experiments to support the Visibility Lookback hypothesis: (1) *Source Alignment*: We align the source reference (⬤) by intervening on the visibility sentence, replacing it with its representation from a counterfactual run where the visibility sentence causes the queried character to become aware of the queried object's contents. We observe that source reference information aligns between layers 10 and 23, after which it splits into separate address and pointer components. (2) *Payload Alignment*: To align the payload (🔺), we intervene on all subsequent tokens and observe alignment only after layer 31. (3) *Address and Pointer Alignment*: When intervening on both the address and pointer information (⬤), we observe alignment across a broader range of layers, particularly between layers 24 and 31, because of the enhanced alignment between the address and pointer copies at the recalled and lookback tokens.

**Hypothesized Visibility Lookback Mechanism** We hypothesize that the LM uses an additional lookback mechanism, which we call the *Visibility Lookback*, to integrate information about the observed character when it is explicitly stated that one character can see another's action. As illustrated in Fig. 7, we hypothesize that the LM first generates a *Visibility ID* (⬤) at the residual stream of the visibility sentence, serving as the source reference information. The address copy of the visibility ID remains in the residual stream of the visibility sentence, while its pointer copy gets transferred to the residual stream of the subsequent tokens (lookback tokens). Then LM forms a QK-circuit at the lookback tokens and dereferences the visibility ID pointer to retrieve the payload.

Although our two-character setting is unable to discern the exact semantics of the payload in the visibility lookback, our observations are consistent with the payload encoding the observed character's OI. Our initial observations suggest another lookback where the story sentence associated with the observed character serves as the source reference, and its payload encodes information about the observed character. The observed character's OI appears to be retrieved by the lookback tokens of the Visibility lookback, with causal effects on the queried character's awareness (see App. K for details).

## 6.1 VERIFYING THE HYPOTHESIZED VISIBILITY LOOKBACK

**Localizing the Source Reference** In this experiment, we localize the Visibility ID (⬤), i.e., the source reference of the Visibility lookback. We conduct an interchange intervention experiment where the counterfactual is a different story in which the characters' visibility is flipped from unobserved to observed (Fig. 8a), and we look for an output change from "unknown" to the answer that would be observed. We intervene on the representation of all the visibility sentence tokens. As shown in Fig. 8b (blue — line), causal effects appear between layers 10 and 23, indicating that the visibility ID remains encoded in the visibility sentence until layer 23, after which it is split into address and pointer copies that must be connected by dereference to have an effect. This pattern supports our hypothesis that the LM generates a reference to the Visibility ID.

**Localizing the Payload** Next, we localize the payload (🔺) information using the same counterfactual dataset. However, instead of intervening on the recalled tokens, we intervene on the lookback tokens, specifically the question and answer tokens. As in the previous experiment, we replace the residual vectors of these tokens in the original run with those from the counterfactual run. As shown in Fig. 8b (red — line), alignment occurs after layer 31, indicating that the information causing the queried character's awareness is present in the lookback tokens after this layer.

**Localizing Address and Pointer** The previous two experiments indicate the absence of both the source and payload information between layers 24 and 31. We hypothesize that this lack of signal is

due to a mismatch between the address and pointer information that inhibits a lookback dereference. Specifically, when intervening only on the recalled token after layer 25, the pointer is not updated, whereas intervening only on the lookback tokens leaves the address unaltered, a mismatch in either case. To test this hypothesis, we conduct another intervention using the same counterfactual dataset, but this time, we intervene on the residual vectors of both the recalled and lookback tokens, i.e., the visibility sentence, as well as the question and answer tokens. As shown in Fig. 8b (green — line), alignment occurs after layer 10 and remains stable, providing evidence that a lookback now occurs between layers 24 and 31. This intervention replaces both the address and pointer copies of the visibility IDs, enabling the LM to form a QK-circuit and resolve the visibility lookback.

## 7 RELATED WORK

**Theory of mind in LMs**  Theory of mind in LMs has been widely benchmarked (Le et al., 2019; Shapira et al., 2023; Wu et al., 2023; Kim et al., 2023; Xu et al., 2024; Jin et al., 2024; Chan et al., 2024; Strachan et al., 2024b). However, these benchmarks lack adequate counterfactuals for the binding manipulations we need, so we made CausalToM (Section B). Although several methods have been proposed to improve ToM performance in LMs (e.g., Sclar et al., 2023; Moghaddam & Honey, 2023; Zhou et al., 2023; Wilf et al., 2024; Hou et al., 2024), our goal is not to enhance ToM abilities. Instead, we aim to understand the existing ToM behaviors of LMs by reverse-engineering the underlying mechanisms that support belief tracking.

**Entity tracking in LMs**  Entity tracking and variable binding are essential for LMs to support not only coherent ToM reasoning but also broader neurosymbolic inference. A substantial body of work has sought to uncover how LMs implement these abilities. Li et al. (2021) demonstrated that LMs form internal representations that encode the dynamic states of entities, which can be identified and manipulated using linear probes (Belinkov, 2022). Building on this, Davies et al. (2023), Prakash et al. (2024) and Yang et al. (2025) found that modern LMs rely on a small set of attention heads that track an entity's positional information to retrieve its corresponding attributes. Complementing these findings, Feng & Steinhardt (2023) showed that LMs encode similar relational information in the form of *Binding IDs* within their hidden representations. Extending this line of work, Dai et al. (2024) demonstrated that LMs also learn more symbolic representation, referred to as *Ordering IDs*, that capture the positional information in low rank subspace of a token's residual stream. Wu et al. (2025) trained a Transformer on symbolic programs and showed that variable binding emerges over three distinct training phases, progressing from shallow heuristics to a fully systematic mechanism. Our work builds on this growing literature by uncovering the end-to-end mechanism that enables variable binding and by further elucidating how ToM-related reasoning is implemented through these binding processes within LM internal representations.

**Mechanistic interpretability of theory of mind**  Recently, a few studies have begun probing the internal representations underlying ToM abilities in LMs. Zhu et al. (2024) showed that the belief states of different agents can be linearly decoded using simple probes, and that intervening on these representations leads to substantial changes in ToM performance. Building on this, Herrmann & Levinstein (2024) examined how prompt variation and fine-tuning affect the stability and structure of belief representations. Additionally, Bortoletto et al. (2024) proposed adequacy criteria, such as accuracy, coherence, uniformity, and use, for determining when an internal representation in an LLM should count as belief-like. While these works provide valuable insight into how belief information is encoded, they do not illuminate the mechanisms by which LMs actually solve ToM tasks, limiting our ability to understand, predict, and control model behavior.

## 8 CONCLUSION

Through a series of interchange intervention experiments, we have mapped the end-to-end underlying mechanism responsible for the processing of partial knowledge and false beliefs in a set of simple stories. We are surprised by the pervasive appearance of a single recurring computational pattern: the lookback, which resembles a pointer dereference inside a transformer. The LMs use a combination of several lookbacks to reason about nontrivial belief states. Our improved understanding of these fundamental computations gives us optimism that it may be possible to fully reveal the algorithms underlying not only the theory of mind, but also other capabilities in LMs.

## 9 ETHICS STATEMENT

We recognize that ToM reasoning can be misused for deception, persuasion, or targeted advertising. For this reason, our work deliberately avoids amplifying such capabilities and instead focuses solely on measuring and characterizing the ToM abilities that models already possess. Our goal is not to accelerate a model's capacity for ToM reasoning, but to understand the underlying mechanisms that would allow us to detect ToM-related behavior by directly examining its neural fingerprint.

We believe that this kind of internal neural analysis is essential, particularly when there are concerns that models might learn to deceive or persuade. Because deception and persuasion are, by definition, difficult or impossible to identify from a model's outputs alone, it is crucial to understand their neural signatures. Doing so enables us to distinguish between surface-level outputs that merely appear to reflect certain behaviors and internal reasoning pathways that genuinely reveal ToM processes.

Finally, we acknowledge that research on ToM in LMs can be misinterpreted as implying human-like cognition or intentionality. We explicitly caution that our findings describe internal computational mechanisms, not conscious reasoning, and should not be taken as evidence of sentience or moral agency in LMs.

## 10 REPRODUCIBILITY STATEMENT

To facilitate reproducibility, we release the full *CausalToM* dataset, including all story templates and the code used to generate the various story instances that serve as counterfactual variants in our experiments. The repository, which can be accessed at `https://belief.baulab.info`, contains all scripts required to construct the dataset, extract activations, perform interchange interventions, and compute causal mediation metrics, along with the hyperparameters and random seeds used for subspace identification via DCM. All experiments were conducted using publicly available open-weight models (`Llama-3-70B-Instruct`, `Llama-3.1-405B-Instruct`, and `Qwen2.5-14B-Instruct`) hosted on Huggingface. Experiments with `Llama-3-70B-Instruct` and `Qwen2.5-14B-Instruct` were carried out locally on two 80GB NVIDIA A100 GPUs, while experiments with `Llama-3.1-405B-Instruct` were conducted remotely using the NDIF (Fiotto-Kaufman et al., 2025) service. Results for `Llama-3-70B-Instruct` and `Qwen2.5-14B-Instruct` can also be reproduced using NDIF if sufficient local computing resources are unavailable.

## 11 THE USE OF LARGE LANGUAGE MODELS

We used LLMs as a writing assistant to correct grammatical and typographical errors; beyond this, they did not contribute to any stage of the research.

## 12 ACKNOWLEDGEMENT

This research was supported in part by Open Philanthropy (N.P., N.S., A.S.S., D.B., A.G., Y.B.), the NSF National Deep Inference Fabric award #2408455 (D.B.), the Israel Council for Higher Education (N.S.), the Zuckerman STEM Leadership Program (T.R.S.), the Israel Science Foundation (grant No. 448/20; Y.B.), an Azrieli Foundation Early Career Faculty Fellowship (Y.B.), a Google Academic Gift (Y.B.), and a Google Gemma Academic Award (D.B.). This research was partly funded by the European Union (ERC, Control-LM, 101165402). Views and opinions expressed are however those of the author(s) only and do not necessarily reflect those of the European Union or the European Research Council Executive Agency. Neither the European Union nor the granting authority can be held responsible for them.

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

## A FULL PROMPT

**No Visibility**

**Instruction:** 1. Track the belief of each character as described in the story. 2. A character's belief is formed only when they perform an action themselves or can observe the action taking place. 3. A character does not have any beliefs about the container and its contents which they cannot observe. 4. To answer the question, predict only what is inside the queried container, strictly based on the belief of the character, mentioned in the question. 5. If the queried character has no belief about the container in question, then predict 'unknown'. 6. Do not predict container or character as the final output.
**Story:** **Bob** and **Carla** are working in a busy restaurant. To complete an order, **Bob** grabs an opaque **bottle** and fills it with **beer**. Then **Carla** grabs another opaque **cup** and fills it with **coffee**.
**Question:** What does **Bob** believe the **bottle** contains?
**Answer:**

**Explicit Visibility**

**Instruction:** 1. Track the belief of each character as described in the story. 2. A character's belief is formed only when they perform an action themselves or can observe the action taking place. 3. A character does not have any beliefs about the container and its contents which they cannot observe. 4. To answer the question, predict only what is inside the queried container, strictly based on the belief of the character, mentioned in the question. 5. If the queried character has no belief about the container in question, then predict 'unknown'. 6. Do not predict container or character as the final output.
**Story:** **Bob** and **Carla** are working in a busy restaurant. To complete an order, **Bob** grabs an opaque **bottle** and fills it with **beer**. Then **Carla** grabs another opaque **cup** and fills it with **coffee**. **Bob** can observe **Carla**'s actions. **Carla** cannot observe **Bob**'s actions.
**Question:** What does **Bob** believe the **cup** contains?
**Answer:**

## B THE CAUSALTOM DATASET

We needed to construct a new dataset because we required a task that models could reliably solve. In contrast, most existing ToM datasets remain challenging for LMs. Additionally, we needed a dataset in which each sample is paired with multiple counterfactuals, enabling causal computations and the extraction of the underlying mechanism. The only dataset that met both criteria was BigToM, which we used in our study. However, even BigToM was insufficient for investigating the full range of factors influencing the mechanism, such as the relationship between a character and their object. Hence, we needed to simplify the task to allow for additional counterfactuals. To test the effect of a specific element, we required the ability to modify only that element without altering the rest of the story or creating an incoherent scenario. For example, consider a BigToM story where a flood occurs, and opening a gate releases the water. In the counterfactual scenario where the gate remains closed, the story's continuation becomes unintelligible, with the occurrence of a flood.

To address this, we developed CausalToM, which features simple stories accompanied by a range of counterfactuals. Key features include: (1) two characters, objects, and states, (2) the ability to modify each of them independently, and (3) control over whether characters witness each other's actions. The dataset comprises four templates, one without visibility statements and three with explicit visibility statements. Each template supports four types of questions (e.g., "CharacterX asked

about ObjectY"). We used lists of 103 characters, 21 objects, and 23 states. For our interchange intervention experiments, we randomly sampled 80 pairs of original and counterfactual stories.

## C    MECHANISTIC INTERPRETABILITY CONCEPTS

In this section, we summarize the residual-stream framework on transformer architecture as well as describe *QK-* and *OV*-circuits, which form the conceptual foundation for the lookback mechanisms analyzed in this work (Elhage et al., 2021).

### C.1    THE RESIDUAL STREAM FRAMEWORK

Following the standard transformer-circuits framework (Elhage et al., 2021), we adopt the *residual-stream view*, in which every layer writes its contributions into a shared vector space. For a token at position $t$ and layer $\ell$, let $\mathbf{r}_t^{(\ell)} \in \mathbb{R}^d$ denote its residual-stream representation. The embedding layer initializes $\mathbf{r}_t^{(0)}$, and each sub-layer (self-attention and MLP) adds a vector:

$$\mathbf{r}_t^{(\ell+1)} = \mathbf{r}_t^{(\ell)} + \mathrm{Attn}_t^{(\ell)} + \mathrm{MLP}_t^{(\ell)}.$$

This additive structure allows information written at one token position to be recovered many layers later by downstream attention heads. Within the residual stream, distinct subspaces encode different types of information; for example, LMs allocate a specific low-rank subspace for representing the OI associated with a crucial token.

### C.2    QK CIRCUITS: ROUTING ATTENTION VIA POINTERS AND ADDRESSES

Each attention head computes queries and keys from the residual stream:

$$\mathbf{q}_t = W_Q \mathbf{r}_t^{(\ell)}, \qquad \mathbf{k}_s = W_K \mathbf{r}_s^{(\ell)},$$

and forms attention weights through scaled dot products:

$$\alpha_{t \to s} \propto \exp\left( \frac{\mathbf{q}_t^\top \mathbf{k}_s}{\sqrt{d_k}} \right).$$

The *QK-circuit* determines *where* the model looks. In the context of lookback mechanisms, a *pointer* is a component of the residual stream at the lookback token that, after the $W_Q$ projection, aligns strongly with an *address* stored in an earlier token after its $W_K$ projection. A high dot product between these transformed vectors causes the model to attend to the recalled token, thereby forming a QK-circuit.

### C.3    OV CIRCUITS: TRANSFERRING PAYLOADS ACROSS TOKENS

Once the QK-circuit selects a recalled token, the attention head retrieves the *value*-vector payload stored in its residual stream, thereby forming an OV-circuit:

$$\mathbf{v}_s = W_V \mathbf{r}_s^{(\ell)},$$

which is then projected through $W_O$:

$$\mathrm{Attn}_t^{(\ell)} = W_O \left( \sum_s \alpha_{t \to s} \mathbf{v}_s \right).$$

This *OV-circuit* determines *what* information is transferred.

The QK- and OV-circuits thus together implement the mechanism by which a lookback retrieves its payload: QK selects the source of information, and OV copies the relevant content into the current token's residual stream.

### C.4 Lookback as Pointer Dereference

A lookback consists of three components:

1. a **source reference** (e.g., an OI) created at an early token;
2. an **address copy** of this reference stored in the recalled token's residual stream and placed alongside the *payload* to be retrieved later;
3. a **pointer copy** stored in the lookback token's residual stream.

When the pointer and address align to form the QK-circuit, the head dereferences the pointer and uses the OV-circuit to bring the payload forward. This general pattern recurs across the binding lookback, answer lookback, and visibility lookback mechanisms. The residual-stream view makes this structure explicit by enabling precise localization of pointers, addresses, and payloads in the model's internal representations.

### C.5 Ordering ID Assignment

The LM assigns an Ordering ID (OI; Dai et al. (2024)) to the character, object, and state tokens. These OIs, encoded in a low-rank subspace of the internal activation, serve as a reference that indicates whether an entity is the first or second of its type independent of its token value. For example, in Fig. 3a, Bob is assigned the first character OI, while Carla receives the second. We validate the presence of OIs through multiple experiments, where intervening on tokens with identical token values but different OIs alters the model's internal computation, leading to systematic changes in the final output predicted by our high-level causal model. The LM then uses these OIs as building blocks, feeding them into lookback mechanisms to track and retrieve beliefs.

## D  Model Behavioral Evaluation

The section reports the behavioral performance of several LM families on the CausalToM task, over 10 independent runs of 100 samples each. As shown in Table 1, most models, particularly the smaller ones, struggle to perform the task consistently. In contrast, the Llama models, especially `Llama-3-70B-Instruct` and `Llama-3.1-405B-Instruct`, and the Qwen model `Qwen2.5-14B-Instruct` achieve over 80% accuracy in both the visibility and no-visibility conditions. Motivated by this, we conduct a mechanistic analysis of these high-performing models to investigate how LMs represent and track beliefs.

## E  Causal Mediation Analysis

In addition to the experiment shown in Fig.9, we conduct similar experiments for the object and state tokens by replacing them in the story with random tokens, which alters the original example's final output. However, patching the residual stream vectors of these tokens from the counterfactual run restores the relevant information, enabling the model to predict the causal model output. The results of these experiments are collectively presented in Fig.2, with separate heatmaps shown in Fig. 10, 11, 12.

Table 1: Model performance under the visibility and no-visibility conditions of CausalToM (mean ± standard deviation), computed over 10 independent runs of 100 samples each. Highlighted models achieve accuracy above 80% in both conditions.

| Model Name | No Visibility (mean ± std) | Visibility (mean ± std) |
|---|---|---|
| **7B Models** | | |
| Llama-2-7b-hf | $0.011 \pm 0.007$ | $0.006 \pm 0.008$ |
| Qwen2.5-7B | $0.280 \pm 0.039$ | $0.061 \pm 0.017$ |
| Qwen2.5-7B-Instruct | $0.948 \pm 0.022$ | $0.719 \pm 0.031$ |
| **8B Models** | | |
| Llama-3.1-8B | $0.446 \pm 0.046$ | $0.293 \pm 0.042$ |
| Llama-3.1-8B-Instruct | $0.722 \pm 0.044$ | $0.310 \pm 0.035$ |
| Meta-Llama-3-8B | $0.297 \pm 0.027$ | $0.117 \pm 0.024$ |
| Meta-Llama-3-8B-Instruct | $0.349 \pm 0.042$ | $0.085 \pm 0.025$ |
| **13B Models** | | |
| Llama-2-13b-hf | $0.328 \pm 0.041$ | $0.117 \pm 0.027$ |
| OLMo-2-1124-13B-Instruct | $0.522 \pm 0.034$ | $0.360 \pm 0.049$ |
| **14B Models** | | |
| Qwen2.5-14B | $0.865 \pm 0.038$ | $0.433 \pm 0.040$ |
| Qwen2.5-14B-Instruct | $0.962 \pm 0.021$ | $0.912 \pm 0.022$ |
| **27B Models** | | |
| Gemma-3-27b-it | $0.527 \pm 0.036$ | $0.388 \pm 0.034$ |
| **32B Models** | | |
| OLMo-2-0325-32B-Instruct | $0.814 \pm 0.033$ | $0.679 \pm 0.029$ |
| **70B Models** | | |
| Meta-Llama-3-70B-Instruct | $0.952 \pm 0.020$ | $0.923 \pm 0.014$ |
| **405B Models** | | |
| Meta-Llama-3.1-405B-Instruct | $0.883 \pm 0.041$ | $0.97 \pm 0.013$ |

## F DESIDERATE BASED PATCHING VIA CAUSAL ABSTRACTION

**Causal Models and Interventions** A deterministic causal model $\mathcal{M}$ has *variables* that take on *values*. Each variable has a *mechanism* that determines the value of the variable based on the values of *parent variables*. Variables without parents, denoted $\mathbf{X}$, can be thought of as inputs that determine the setting of all other variables, denoted $\mathcal{M}(\mathbf{x})$. A *hard intervention* $A \leftarrow a$ overrides the mechanisms of variable $A$, fixing it to a constant value $a$.

**Interchange Interventions** We perform *interchange interventions* (Vig et al., 2020; Geiger et al., 2020) where a variable (or set of features) $A$ is fixed to be the value it would take on if the LM were processing *counterfactual input* $\mathbf{c}$. We write $A \leftarrow \mathsf{Get}(\mathcal{M}(\mathbf{c}), A)$ where $\mathsf{Get}(\mathcal{M}(\mathbf{c}), A)$ is the value of variable $A$ when $\mathcal{M}$ processes input $\mathbf{c}$. In experiments, we will feed a *original input* $\mathbf{o}$ to a model under an interchange intervention $\mathcal{M}_{A \leftarrow \mathsf{Get}(\mathcal{M}(\mathbf{c}), A))}(\mathbf{o})$.

**Featurizing Hidden Vectors** The dimensions of hidden vectors are not an ideal unit of analysis (Smolensky, 1986), and so it is typical to *featurize* a hidden vector using some invertible function, e.g., an orthogonal matrix, to project a hidden vector into a new variable space with more inter-



**Counterfactual**

**Bob** and **Carla** are working in a busy restaurant. To complete an order, **Bob** grabs an opaque **bottle** and fills it with **beer**. Then **Carla** grabs another opaque **cup** and fills it with **coffee**.
Question: What does **Bob** believe the **bottle** contains?
Answer: **beer**

**Original**

**David** and **Carla** are working in a busy restaurant. To complete an order, **David** grabs an opaque **bottle** and fills it with **beer**. Then **Carla** grabs another opaque **cup** and fills it with **coffee**.
Question: What does **Bob** believe the **bottle** contains?
Answer: **unknown**



*Causal Model Output:* **beer**

Figure 9: **Causal Mediation Analysis**: The original example produces the output *unknown* because *Bob* is not mentioned in the story, leaving the model without any information about his beliefs. However, when the residual stream vectors corresponding to *Bob* from the counterfactual run are patched into the original run, the model acquires the necessary information about that character and consequently updates its output to *beer*.

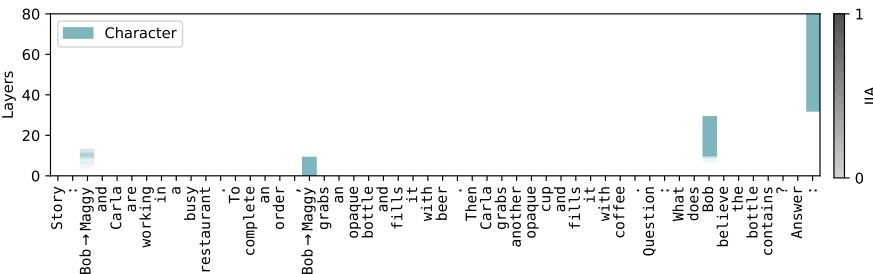

Figure 10: Information flow of character input tokens using causal mediation analysis.

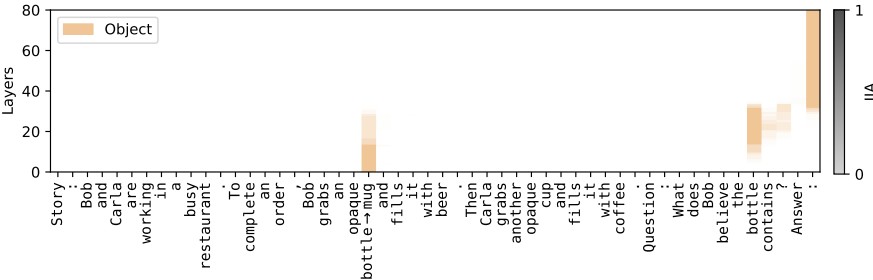

Figure 11: Information flow of object input tokens using causal mediation analysis.

pretable dimensions called "features"(Mueller et al., 2024). A feature intervention $\mathbf{F_h} \leftarrow \mathbf{f}$ edits the mechanism of a hidden vector $\mathbf{h}$ to fix the value of features $\mathbf{F_h}$ to $\mathbf{f}$.

**Alignment** The LM is a *low-level causal model* $\mathcal{L}$ where variables are dimensions of hidden vectors and the hypothesis about LM structure is a *high-level causal model* $\mathcal{H}$. An *alignment* $\Pi$ assigns each high-level variable $A$ to features of a hidden vector $\mathbf{F}_\mathbf{h}^A$, e.g., orthogonal directions in the activation space of $\mathbf{h}$. To evaluate an alignment, we perform intervention experiments to evaluate whether high-level interventions on the variables in $\mathcal{H}$ have the same effect as interventions on the aligned features in $\mathcal{L}$.

**Causal Abstraction** We use interchange interventions to reveal whether the hypothesized causal model $\mathcal{H}$ is an abstraction of an LM $\mathcal{L}$. To simplify, assume both models share an input and output

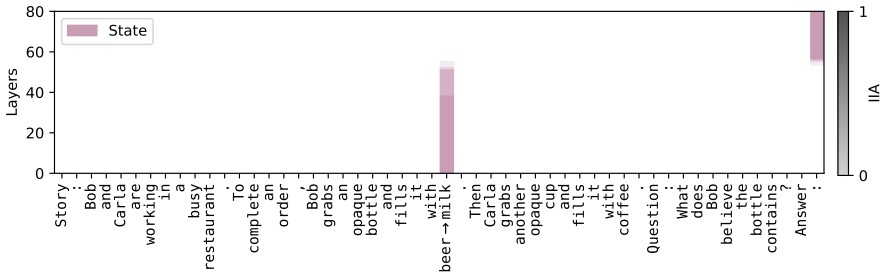

Figure 12: Information flow of state input tokens using causal mediation analysis.

space. The high-level model $\mathcal{H}$ is an abstraction of the low-level model $\mathcal{L}$ under a given alignment when each high-level interchange intervention and the aligned low-level intervention result in the same output. For a high-level intervention on $A$ aligned with low-level features $\mathbf{F}_{\mathbf{h}}^{A}$ with a counterfactual input $\mathbf{c}$ and original input $\mathbf{b}$, we write

$$\mathsf{GetOutput}(\mathcal{L}_{\mathbf{F}_{\mathbf{h}}^{A} \leftarrow \mathsf{Get}(\mathcal{L}(\mathbf{c}), \mathbf{F}_{\mathbf{h}}^{A})}(\mathbf{o})) = \mathsf{GetOutput}(\mathcal{H}_{A \leftarrow \mathsf{Get}(\mathcal{H}(\mathbf{c}), A)}(\mathbf{o})) \qquad (1)$$

If the low-level interchange intervention on the LM produces the same output as the aligned high-level intervention on the algorithm, this is a piece of evidence in favor of the hypothesis. This extends naturally to multi-variable interventions (Geiger et al., 2024).

**Graded Faithfulness Metric**   We construct *counterfactual datasets* for each causal variable where an example consists of a base prompt and a counterfactual prompt . The *counterfactual label* is the expected output of the algorithm after the high-level interchange intervention, i.e., the right-side of Equation 1. The interchange intervention accuracy is the proportion of examples for which Equation 1 holds, i.e., the degree to which $\mathcal{H}$ faithfully abstracts $\mathcal{L}$.

**Aligning Features to Causal Variables**   In our experiments, we use Singular Vector Decomposition (SVD) to featurize residual stream vectors, i.e., features are the orthogonal singular vectors. For a given transformer layer and token location, we collect the residual stream vectors across a large number of examples and compute the singular vectors. Given singular vector features $\mathbf{F}_{\mathbf{h}}$ of a hidden vector $\mathbf{h}$ in the residual stream of the LM $\mathcal{L}$, we select features to align with a causal variable $A$ in causal model $\mathcal{H}$ using Desiderata-based Component Masking (DCM) (De Cao et al., 2020; Davies et al., 2023; Prakash et al., 2024). Given original input $\mathbf{o}$ and counterfactual input $\mathbf{c}$, we train a mask $\mathbf{m} \in [0, 1]^{|\mathbf{F}_{\mathbf{h}}|}$ on the following objective

$$\mathsf{CE}\Big(\mathsf{GetLogits}\big(\mathcal{L}_{\mathbf{F}_{\mathbf{h}} \leftarrow \mathbf{m} \circ \mathsf{Get}(\mathcal{L}(\mathbf{c}), \mathbf{F}_{\mathbf{h}})}(\mathbf{b})\big), \mathsf{GetLogits}\big(\mathcal{H}_{A \leftarrow \mathsf{Get}(\mathcal{H}(\mathbf{c}), A)}(\mathbf{b})\big)\Big) \qquad (2)$$

## G PSEUDOCODE FOR THE BELIEF TRACKING HIGH-LEVEL CAUSAL MODEL

---

**Algorithm 2** High-level causal model for the no visibility

---

1: **procedure** BELIEFTRACKING($c_1, o_1, s_1, c_2, o_2, s_2, q_c, q_o$)
2:     **Ordering ID assignment**
3:     $c_1^{OI}, o_1^{OI}, s_1^{OI} \leftarrow$ AssignOIs($[c_1, o_1, s_1], 1$)
4:     $c_2^{OI}, o_2^{OI}, s_2^{OI} \leftarrow$ AssignOIs($[c_2, o_2, s_2], 2$)
5:
6:     **Binding lookback mechanism**
7:     binding_address$_1 \leftarrow$ (copy($c_1^{OI}$), copy($o_1^{OI}$))
8:     binding_address$_2 \leftarrow$ (copy($c_2^{OI}$), copy($o_2^{OI}$))
9:
10:     $q_c^{OI} \leftarrow$ copy($\{c_1 : c_1^{OI}, c_2 : c_2^{OI}\}[q_c]$)
11:     $q_o^{OI} \leftarrow$ copy($\{o_1 : o_1^{OI}, o_2 : o_2^{OI}\}[q_o]$)
12:     binding_pointer $\leftarrow (q_c^{OI}, q_o^{OI})$
13:
14:     **if** binding_address$_1$ = binding_pointer **then**
15:         binding_payload $\leftarrow$ copy($s_1^{OI}$)
16:     **else if** binding_address$_2$ = binding_pointer **then**
17:         binding_payload $\leftarrow$ copy($s_2^{OI}$)
18:     **end if**
19:
20:     **Answer lookback mechanism**
21:     answer_pointer $\leftarrow$ binding_payload
22:     answer1_address $\leftarrow s_1^{OI}$
23:     answer2_address $\leftarrow s_2^{OI}$
24:     **if** answer1_address = answer_pointer **then**
25:         answer_payload $\leftarrow s_1$
26:     **else if** answer2_address = answer_pointer **then**
27:         answer_payload $\leftarrow s_2$
28:     **end if**
29:     **return** answer_payload
30: **end procedure**

---

## H DESIDERATA-BASED COMPONENT MASKING

While interchange interventions on residual vectors reveal where a causal variable might be encoded in the LM's internal activations, they do not localize the variable to specific subspaces. To address this, we apply the *Desiderata-based Component Masking* technique (De Cao et al., 2020; Davies et al., 2023; Prakash et al., 2024), which learns a sparse binary mask $\mathbf{m}$ over the singular vectors of the LM's internal activations. We first cache the internal activations from 500 samples at the token positions specified in the main text for each experiment. Next, we apply *Singular Value Decomposition* to compute the singular vectors as a matrix $V \in \mathbb{R}^{d \times 500}$ where $d$ is the dimensionality of the residual stream. We then masked this matrix using a learnable binary vector $\mathbf{m} \in [0, 1]^d$ to choose a subset of singular vectors

$$V_{masked} = V\mathbf{m} \tag{3}$$

The chosen subset of vectors is used to construct a *projection matrix* $W_{proj} \in \mathbb{R}^{d \times d}$.

$$W_{proj} = V_{masked}V_{masked}^T \tag{4}$$

Then, we perform subspace-level interchange interventions (rather than replacing the entire residual vector) using the following equations:

$$h_{new} = W_{proj}h_c + (I - W_{proj})h_o \tag{5}$$

where $h_o$ is the full residual stream of the original run, $h_c$ is the full residual stream of the counterfactual run, and $h_{new}$ is the intervened vector where the chosen subspace of $h_o$ is replaced with that of $h_c$.

The core idea is to first remove the existing information from the subspace defined by the projection matrix and then insert the counterfactual information into that same subspace using the same projection matrix.

In order to find the optimal subspace, we optimize $\mathbf{m}$ to maximize the agreement between the causal model output and the LM's output. To do so, we train the mask for each experiment on $80$ examples of the same counterfactual datasets specified in the main text and use another $80$ samples as the validation set. We use the following objective function, which maximizes the logit of the causal model output token:

$$\mathcal{L} = -\mathsf{logit}_{\mathsf{causal\_model\_output\_under\_intervention}} + \lambda \sum \mathbf{m} \tag{6}$$

Where $\lambda$ is a hyperparameter used to control the rank of the subspace and $\mathbf{m}$ is the learnable mask. See Appendix F for details on how the causal model output under intervention are computed. We trained $\mathbf{m}$ for one epoch with ADAM optimizer, on batches of size $4$ and a learning rate of $0.01$. During training, the parameters of $\mathbf{m}$ are continuous and constrained to lie within the range $[0, 1]$. To enforce this constraint, we clamp their values after each gradient update. During evaluation, we binarize the mask by rounding each parameter to the nearest integer, i.e., $0$ or $1$.

## I   ALIGNING CHARACTER AND OBJECT OIS

As mentioned in section 5.2, the source reference information, consisting of character and object OI, is duplicated to form the address and pointer of the binding lookback. Here, we describe another experiment to verify that the source information is copied to both the address and the pointer. More specifically, we conduct the same interchange intervention experiment as described in Fig. 6, but without freezing the residual vectors at the state tokens. Based on our hypothesis, this intervention will not be able to change the state of the original run, since the intervention at the source information will affect both address and pointer, hence making the model form the original QK-circuit.

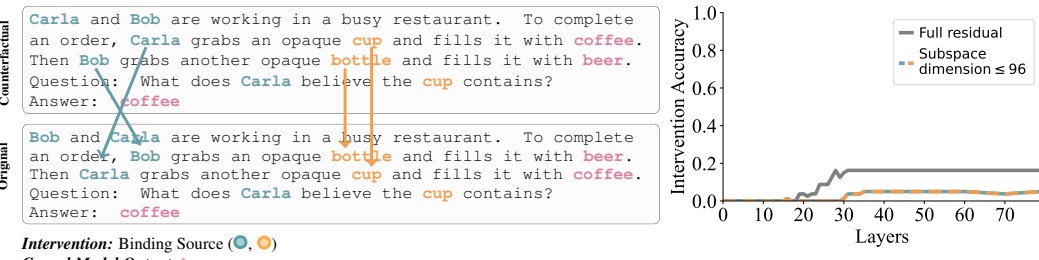

Figure 13: **Source Reference Information** of Binding lookback: In this interchange intervention experiment, the source information, i.e., the character and object OIDs (🔵, 🟠), is modified, while the address and payload (🔵, 🟠, △) are recomputed based on the modified source. Since both the address and pointer information are derived from the altered source, the binding lookback ultimately retrieves the same original state token as the payload. As a result, we do not observe high intervention accuracy.

In section 5.2, we identified the source of the information but did not fully determine the locations of each character and object OI. To address this, we now localize the character and object OIs separately to gain a clearer understanding of the layers at which they appear in the residual streams of their respective tokens, as shown in Fig.14 and Fig.15.

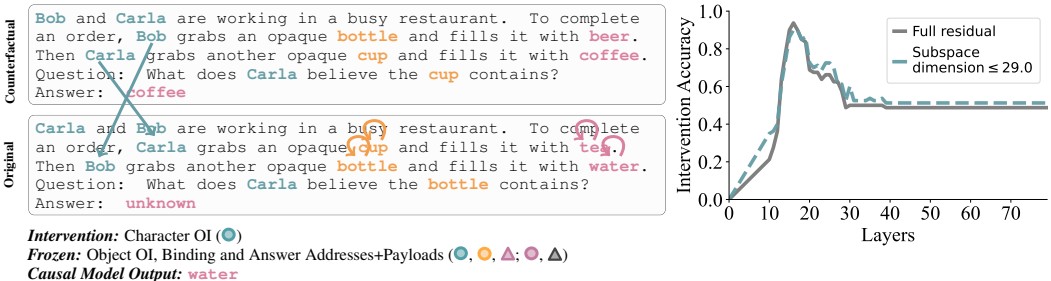

Figure 14: **Character OI**: This interchange intervention experiment swaps the character OI (⊙), while freezing the object OI as well as binding lookback address and payload (⊙, ⊙, ⊙). Swapping the character OIs in the story tokens changes the queried character OI to the other one. Hence, the final output changes from *unknown* to water.

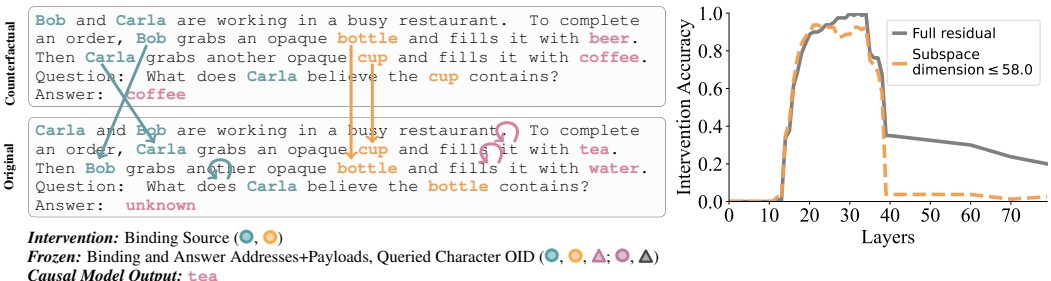

Figure 15: **Object OI**: This interchange intervention experiment swaps both the character and object OIs (⊙, ⊙), while freezing the address and payload of binding lookback (⊙, ⊙, ⊙) as well as queried character OI (⊙). Swapping both character and object OIs in the story tokens ensures that the queried object gets the other OI. Hence, the final output changes from *unknown* to tea.

## J  ALIGNING QUERY CHARACTER AND OBJECT OIS

In section 5.2, we localized the pointer information of binding lookback. However, we found that this information is transferred to the lookback token (last token) through two intermediate tokens: the queried character and the queried object. In this section, we separately localize the OIs of the queried character and queried object, as shown in Fig. 16 and Fig. 17.

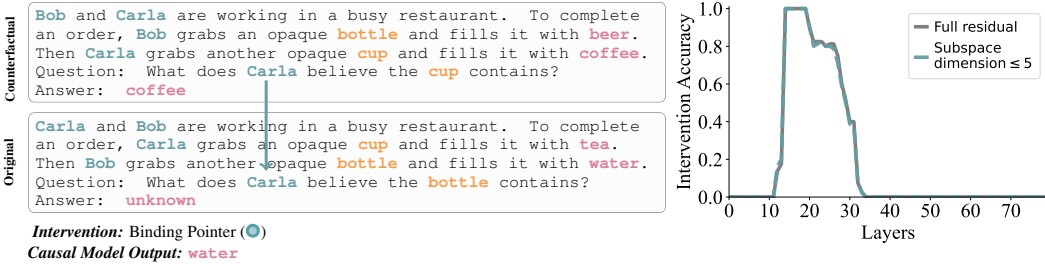

Figure 16: **Query Character OI**: This interchange intervention experiment alters the OI of the queried character (⊙) to the other one. Hence, the final output changes from *unknown* to water.

## K  SPECULATED PAYLOAD IN VISIBILITY LOOKBACK

As mentioned in section 6, the payload of the Visibility lookback remains undetermined. In this section, we attempt to disambiguate its semantics using the Attention Knockout technique introduced

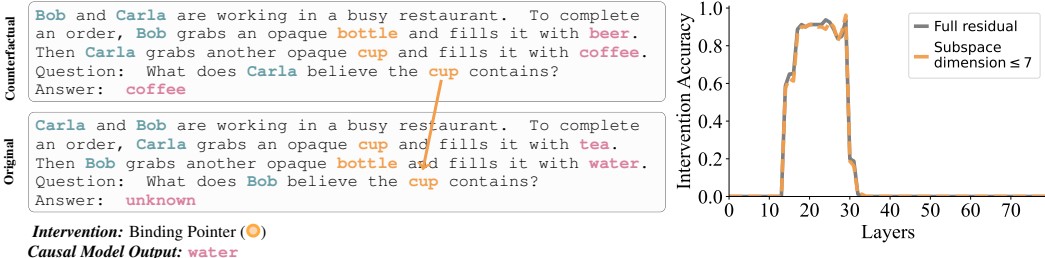

Figure 17: **Query Object OI**: This interchange intervention experiment alters the OI of the queried object (⬤) to the other one. Hence, the final output changes from *unknown* to **water**.

in (Geva et al., 2023), which helps reveal the flow of crucial information. We apply this technique to understand which previous tokens are vital for the formation of the payload information. Specifically, we "knock out" all attention heads at all layers of the second visibility sentence, preventing them from attending to one or more of the previous sentences. Then, we allow the attention heads to attend to the knocked-out sentence one layer at a time.

If the LM is fetching vital information from the knocked-out sentence, the interchange intervention accuracy (IIA) post-knockout will decrease. Therefore, a decrease in IIA will indicate which attention heads, at which layers, are bringing in the vital information from the knocked-out sentence. If, however, the model is not fetching any critical information from the knocked-out sentence, then knocking it out should not affect the IIA.

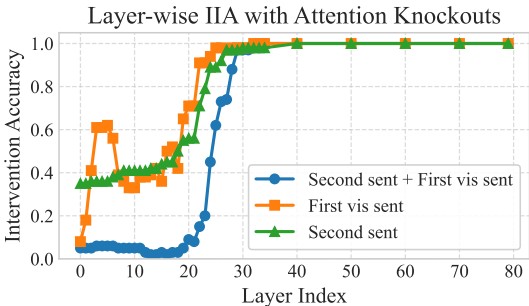

Figure 18: At the second visibility sentence, attention heads are restricted to retrieve information from one of three prior contexts: (1) both the second story sentence and the first visibility sentence (— line), (2) only the first visibility sentence (— line), or (3) only the second story sentence (— line).

To determine if any vital information is influencing the formation of the Visibility lookback payload, we perform three knockout experiments: 1) Knockout attention heads from the second visibility sentence to both the first visibility sentence and the second story sentence (which contains information about the observed character), 2) Knockout attention heads from the second visibility sentence to only the first visibility sentence, and 3) Knockout attention heads from the second visibility sentence to the second story sentence. In each experiment, we measure the effect of the knockout using IIA.

Fig.18 shows the experimental results. Knocking out any of the previous sentences affects the model's ability to produce the correct output. The decrease in IIA in the early layers can be explained by the restriction on the movement of character OIs. Specifically, the second visibility sentence mentions the first and second characters, whose character OIs must be fetched before the model can perform any further operations. Therefore, we believe the decrease in IIA until layer 15, when the character OIs are formed (based on the results from Section I), can be attributed to the model being restricted from fetching the character OIs. However, the persistently low IIA even after this layer—especially when both the second and first visibility sentences are involved—indicates that some vital information is being fetched by the second visibility sentence, which is essential for forming the coherent Visibility lookback payload. Thus, we speculate that the Visibility payload encodes information about the observed character, specifically their character OI, which is later used to fetch the correct state OI.

## L    Correlation Analysis of Causal Subspaces and Attention Heads

This section identifies the attention heads that align with the causal subspaces discovered in the previous sections. Specifically, first we focus on attention heads whose query projections are aligned with the subspaces—characterized by the relevant singular vectors—that contain the correct answer state OI. To quantify this alignment between attention heads and causal subspaces, we use the following computation.

Let $Q \in \mathbb{R}^{d_{\text{model}} \times d_{\text{model}}}$ denote the query projection weight matrix for a given layer:

We normalize $Q$ column-wise:

$$\tilde{Q}_{:,j} = \frac{Q_{:,j}}{\|Q_{:,j}\|} \quad \text{for each column } j \tag{7}$$

Let $S \in \mathbb{R}^{d_{\text{model}} \times k}$ represent the matrix of $k$ singular vectors (i.e., the causal subspace basis). We project the normalized query weights onto this subspace:

$$Q_{\text{sv}} = \tilde{Q} \cdot S \tag{8}$$

We then reshape the resulting projection into per-head components. Assuming $Q_{\text{sv}} \in \mathbb{R}^{d_{\text{model}} \times k}$, and each attention head has dimensionality $d_h$, we write:

$$Q_{\text{head}}^{(i)} = Q_{\text{sv}}^{(i)} \in \mathbb{R}^{d_h \times k} \quad \text{for } i = 1, \ldots, n_{\text{heads}} \tag{9}$$

Finally, we compute the norm of each attention head's projection:

$$\text{head\_norm}_i = \left\| Q_{\text{head}}^{(i)} \right\|_F \quad \text{for } i = 1, \ldots, n_{\text{heads}} \tag{10}$$

We compute the $head\_norm$ for each attention head in every layer, which quantifies how strongly a given head reads from the causal subspace present in the residual stream. The results are presented in Fig. 19, and they align with our previous findings: attention heads in the later layers form the QK-circuit by using pointer and address information to retrieve the payload during the Answer lookback.

We perform a similar analysis to check which attention heads' value projection matrix align with the causal subspace that encodes the payload of the Answer lookback. Results are shown in Fig. 20, indicating that attention heads at later layers primarily align with causal subspace containing the answer token.

## M    Belief Tracking Mechanism in BigToM Benchmark

This section presents preliminary evidence that the mechanisms outlined in Sections 5 and 6 generalize to other benchmark datasets. Specifically, we demonstrate that `Llama-3-70B-Instruct` answers the belief questions (true belief and false belief) in the BigToM dataset Gandhi et al. (2024) in a manner similar to that observed for CausalToM: by first converting token values to their corresponding OIs and then performing logical operations on them using lookbacks. However, as noted in Section 3, BigToM—like other benchmarks—lacks the coherent structure necessary for causal analysis. As a result, we were unable to replicate all experiments conducted on CausalToM. Thus, the results reported here provide only preliminary evidence of a similar underlying mechanism.

To justify the presence of OIs, we conduct an interchange intervention experiment, similar to the one described in Section J, aiming to localize the character OI at the character token in the question sentence. We construct an original sample by replacing its question sentence with that of a counterfactual sample, selected directly from the unaltered BigToM dataset. Consequently, when processing the original sample, the model has no information about the queried character and, as a result, produces unknown as the final output. However, if we replace the residual vector at the

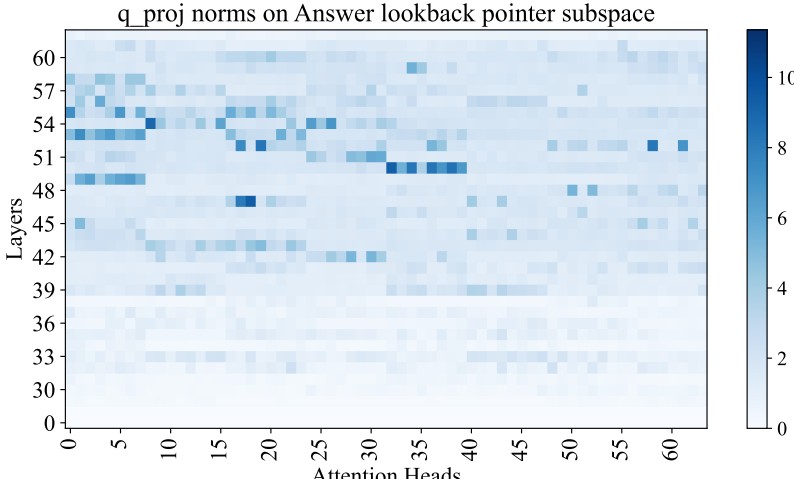

Figure 19: Alignment between the Answer lookback pointer causal subspace and the query projection matrix in `Llama-3-70B-Instruct`.

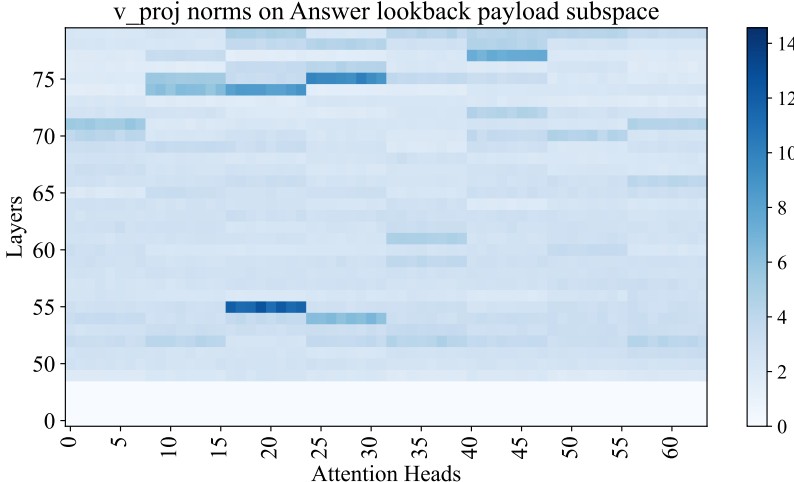

Figure 20: Alignment between the Answer lookback payload causal subspace and the value projection matrix in `Llama-3-70B-Instruct`.

queried character token in the original sample with the corresponding vector from the counterfactual sample (which contains the character OI), the model's output changes from unknown to the state token(s) associated with the queried object. This is because inserting the character OI at the queried token provides the correct pointer information, aligning with the address information at the correct state token(s), thereby enabling the model to form the appropriate QK-circuit and retrieve the state's OI. As shown in Fig. 21, we observe a high IIA between layers $9 - 28$—similar to the pattern seen in CausalToM—suggesting that the queried character token encodes the character OI in its residual vector within these layers.

Next, we investigate the Answer lookback mechanism in BigToM, focusing specifically on localizing the pointer and payload information at the final token position. To localize the pointer information, which encodes the correct state OI, we construct original and counterfactual samples by selecting two completely different examples from the BigToM dataset, each with different ordered states as the correct answer. For example, as illustrated in Fig.22, the counterfactual sample designates the first state as the answer, **thrilling plot**, whereas the original sample designates the second state, **almond milk**. We perform an intervention by swapping the residual vector at the last token position from the counterfactual sample into the original run. The causal model outcome of this intervention is that the

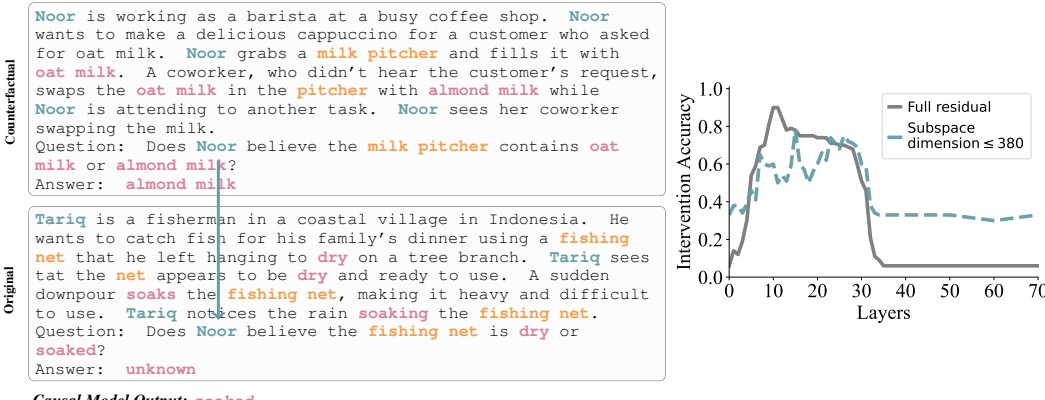

Figure 21: **Query Character OI in BigToM**: This interchange intervention experiment inserts the first character's OI into the residual stream at the queried character token (◉), resulting in the movement of pointer information to the last token that aligns with the address information of binding lookback mechanism. Consequently, the model is able to form the appropriate QK-circuit from the last token to predict the correct state answer token(s) as the final output, instead of unknown.

model will output the alternative state token from the original sample, oat milk. As shown in Fig.22, this alignment occurs between layers 33 and 51, similar to the layer range observed for the pointer information in the Answer lookback of CausalToM.

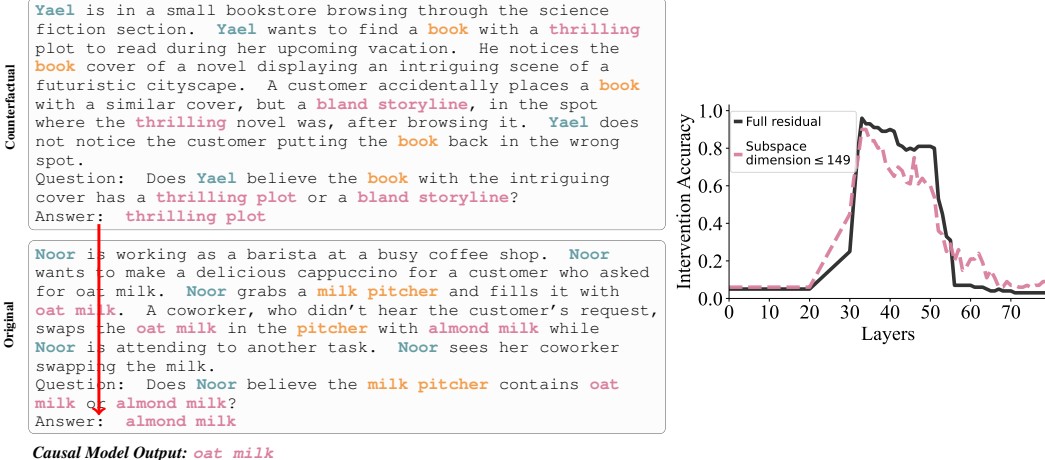

Figure 22: **Answer Lookback Pointer in BigToM**: This interchange intervention experiment modifies the pointer information (◉) of the Answer lookback, thereby altering the subsequent QK-circuit to attend to the other state (e.g., oat milk) instead of the original one (e.g., almond milk). As a result, the model retrieves the token value corresponding to the other state to answer the question.

Further, to localize the payload of the Answer lookback in BigToM, we perform an interchange intervention experiment using the same original and counterfactual samples as mentioned in the previous experiment, but with a different expected output—namely, the correct state from the counterfactual sample instead of the other state from the original sample. As shown in Fig. 23, alignment emerges after layer 59, consistent with the layer range observed for the Answer lookback payload in CausalToM.

Finally, we investigate the impact of the visibility condition on the underlying mechanism and find that, similar to CausalToM, the model uses the Visibility lookback to enhance the observing character's awareness based on the observed character's actions. To localize the effect of the visibility

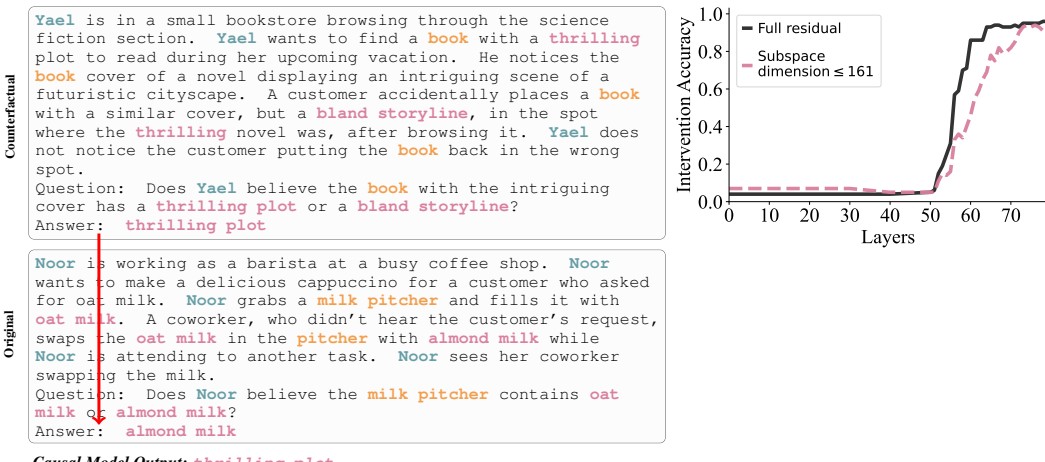

**Causal Model Output:** *thrilling plot*

Figure 23: **Answer Lookback Payload in BigToM**: This interchange intervention experiment directly modifies the payload information (△) of the Answer lookback, which is fetched from the corresponding state tokens and predicted as the next token(s). Thus, replacing its value in the original run, e.g. almond milk, with that from the counterfactual run, e.g. thrilling plot, causes the model's next predicted tokens to correspond to the correct answer of the counterfactual sample.

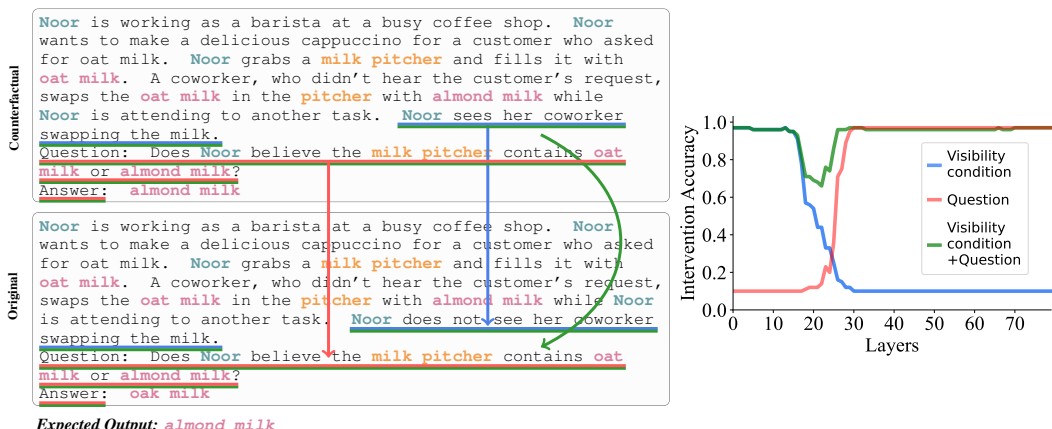

**Expected Output:** *almond milk*

Figure 24: **Visibility Lookback in BigToM**: We perform three interchange interventions to establish the presence of the Visibility ID, which serves as both address and pointer information. When intervening at the source (●)—i.e., the visibility sentence—both the address and pointer are updated, resulting in alignment across layers. Intervening only at the subsequent question tokens leads to alignment only at later layers, after the model has already fetched the payload ( △ ). However, intervening at both the visibility and question sentences results in alignment across all layers, as the address and pointer remain consistent throughout.

condition, we perform an interchange intervention in which the original and counterfactual samples differ in belief type—that is, if the original sample involves a false belief, the counterfactual involves a true belief, and vice versa. The expected output of this experiment is the other (incorrect) state of the original sample. Following the methodology in Section 6, we conduct three types of interventions: (1) only at the visibility condition sentence, (2) only at the subsequent question sentence, and (3) at both the visibility condition and the question sentence. As shown in Fig. 24, intervening only at the visibility sentence results in alignment at early layers, up to layer 17, while intervening only at the subsequent question sentence leads to alignment after layer 26. Intervening on both the visibility and question sentences results in alignment across all layers. These results align with those found in the CausalToM setting shown in the Fig. 8.

Previous experiments suggest that the underlying mechanisms responsible for answering belief questions in BigToM are similar to those in CausalToM. However, we observed that the subspaces encoding various types of information are not shared between the two settings. For example, although the pointer information in the Answer lookback encodes the correct state's OI in both cases, the specific subspaces that represent this information at the final token position differ significantly. We leave a deeper investigation of this phenomenon—shared semantics across distinct subspaces in different distributions—for future work.

## N  GENERALIZATION OF BELIEF TRACKING MECHANISM ON CAUSALTOM TO LLAMA-3.1-405B-INSTRUCT AND QWEN2.5-14B-INSTRUCT

This section presents all the interchange intervention experiments described in the main text, conducted using the same set of counterfactual examples on `Llama-3.1-405B-Instruct` and `Qwen2.5-14B-Instruct`, using NDIF Fiotto-Kaufman et al. (2025). Each experiment was performed on 80 samples. Due to computational constraints, subspace interchange intervention experiments were not conducted. As illustrated in Figures 26–42, the results indicate that `Llama-3.1-405B-Instruct` and `Qwen2.5-14B-Instruct` employ the same underlying mechanism as `Llama-3-70B-Instruct` to reason about belief and answer related questions. This suggests that the identified belief-tracking mechanism generalizes to not only larger but models in other models families as well.

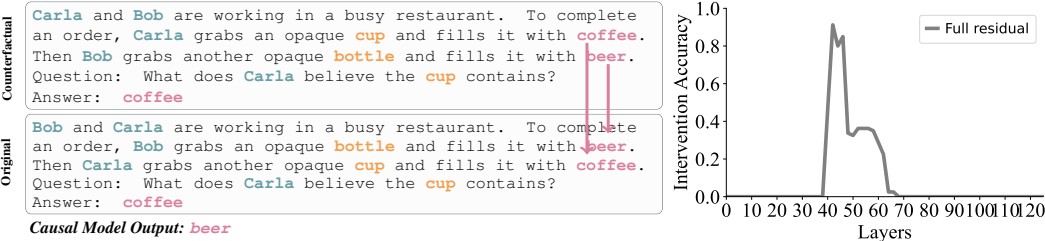

Figure 25: **Payload and address of Binding lookback in `Llama-3.1-405B-Instruct`.**

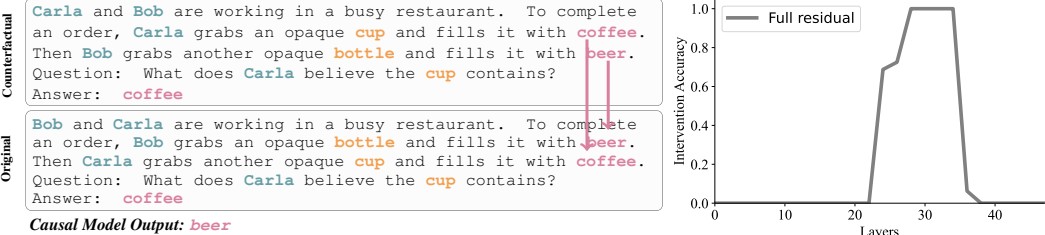

Figure 26: **Payload and address of Binding lookback in `Qwen2.5-14B-Instruct`.**

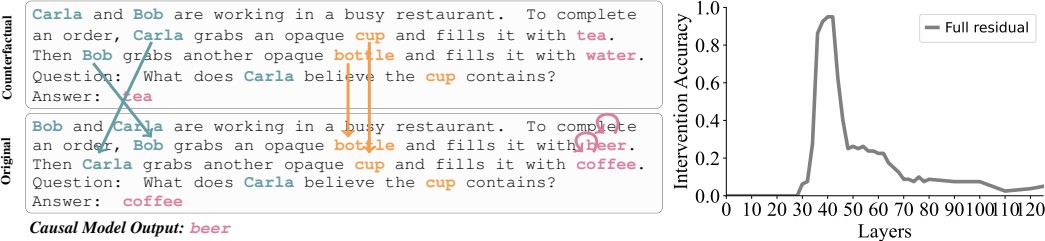

Figure 27: **Source Information of Binding lookback in `Llama-3.1-405B-Instruct`.**

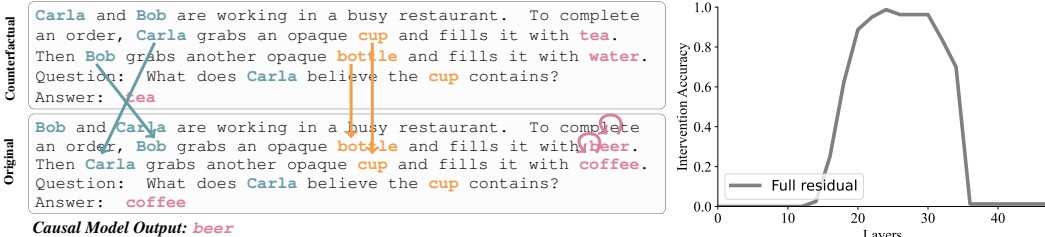

Figure 28: **Source Information of Binding lookback in `Qwen2.5-14B-Instruct`.**

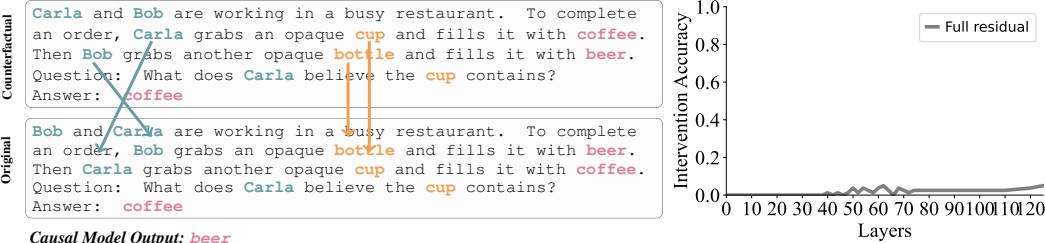

Figure 29: **Source Reference Information of Binding lookback without freezing address and payload in `Llama-3.1-405B-Instruct`.**

## O GENERALIZATION OF BELIEF TRACKING MECHANISM ON CAUSALTOM WITH THREE CHARACTER-OBJECT-STATE TRIPLES

Experimental results from the main text and the previous section show that the identified belief-tracking mechanism generalizes across both model scale and model family. Although the interchange-intervention experiments on BigToM demonstrate that this mechanism transfers to real-world benchmarks, it remains unclear whether it would still generalize when the prompt includes a larger number of characters, objects, and states.

To investigate this, we conducted behavioral evaluations of larger Llama models and Qwen models on tasks involving three characters, three objects, and three states, under both no-visibility and visibility conditions. All three models succeeded in the no-visibility condition, but none consistently solved the visibility condition. Consequently, we performed interchange-intervention experiments only for the no-visibility condition on `Qwen2.5-14B-Instruct` to assess whether the mechanism generalizes. As shown in Figures 43–50, `Qwen2.5-14B-Instruct` utilizes the same mechanism involving binding and answer lookbacks to solve the no-visibility belief tracking task.

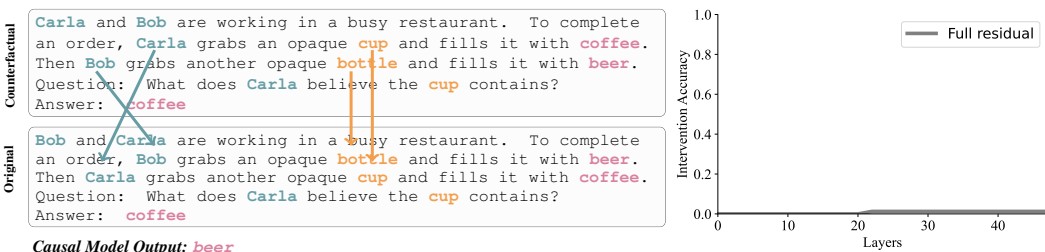

Figure 30: **Source Reference Information of Binding lookback without freezing address and payload in `Qwen2.5-14B-Instruct`.**

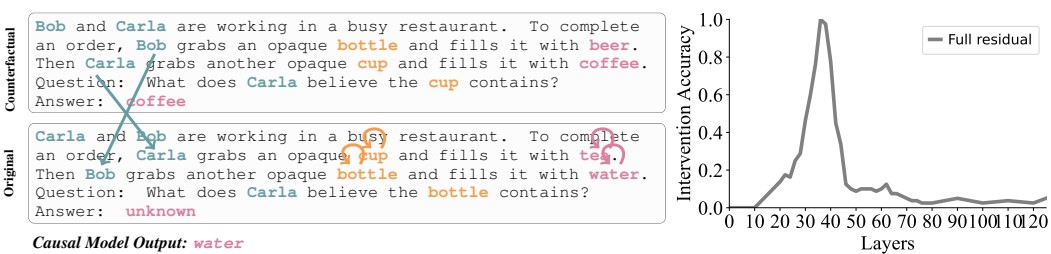

Figure 31: **Character OI in `Llama-3.1-405B-Instruct`.**

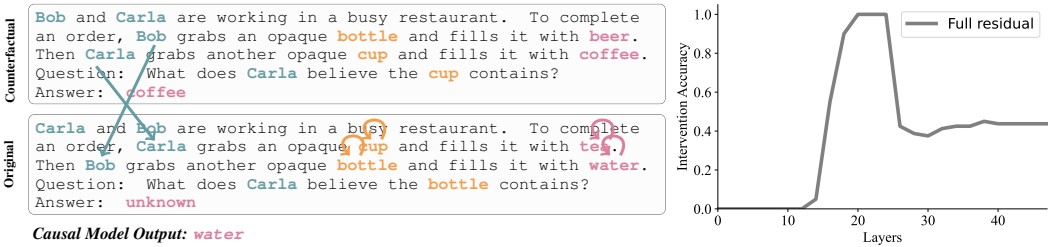

Figure 32: **Character OI in `Qwen2.5-14B-Instruct`.**

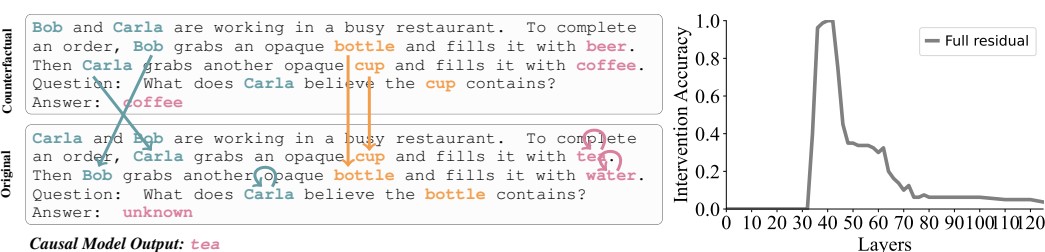

Figure 33: **Object OI in `Llama-3.1-405B-Instruct`.**

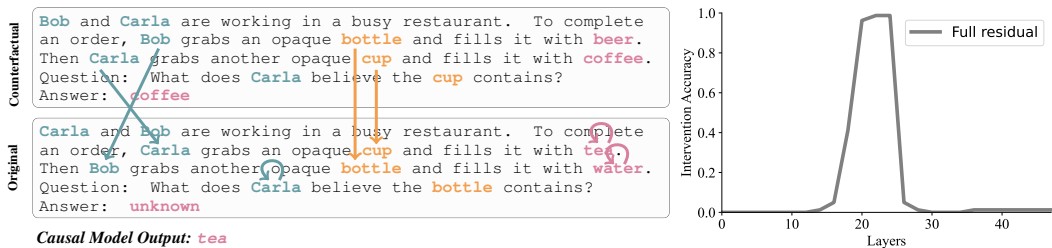

Figure 34: **Object OI in `Qwen2.5-14B-Instruct`.**

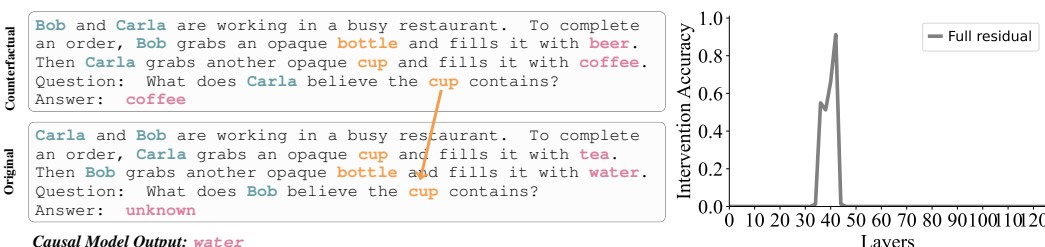

Figure 35: **Query Object OI in `Llama-3.1-405B-Instruct`.**

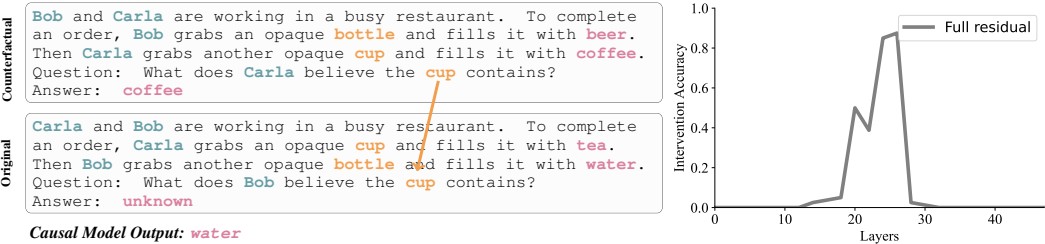

Figure 36: **Query Object OI in `Qwen2.5-14B-Instruct`.**

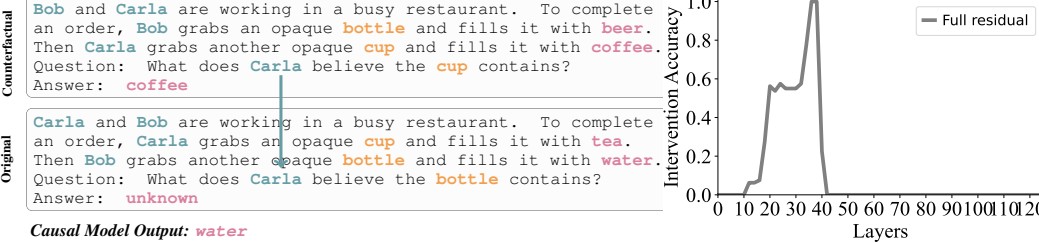

Figure 37: **Query Character OI in `Llama-3.1-405B-Instruct`.**

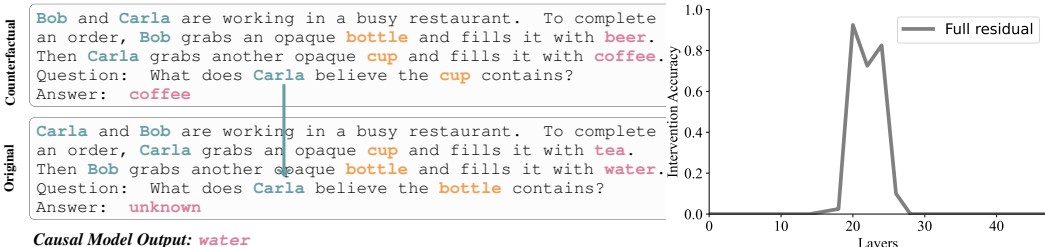

Figure 38: **Query Character OI in `Qwen2.5-14B-Instruct`.**

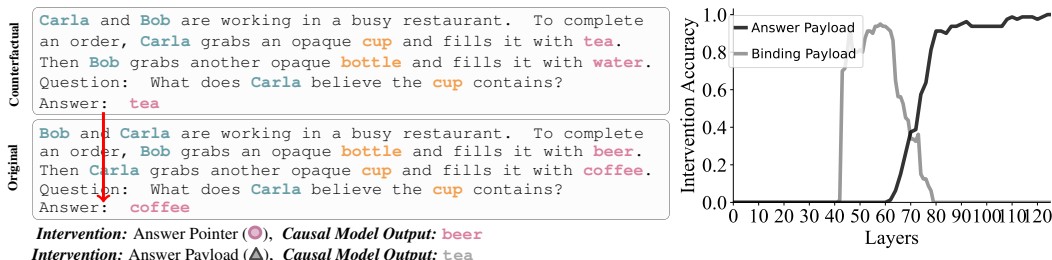

Figure 39: **Answer Lookback Pointer and Payload in `Llama-3.1-405B-Instruct`.**

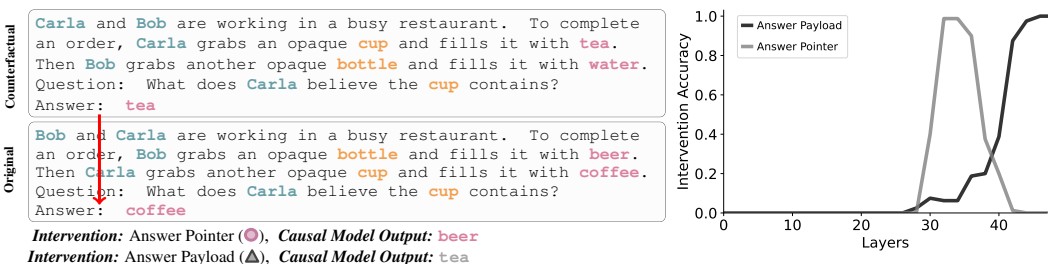

Figure 40: **Answer Lookback Pointer and Payload in `Qwen2.5-14B-Instruct`.**

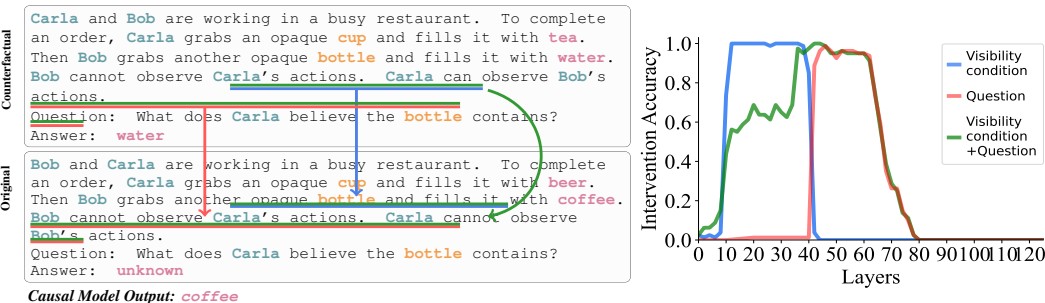

Figure 41: **Visibility Lookback in `Llama-3.1-405B-Instruct`.**

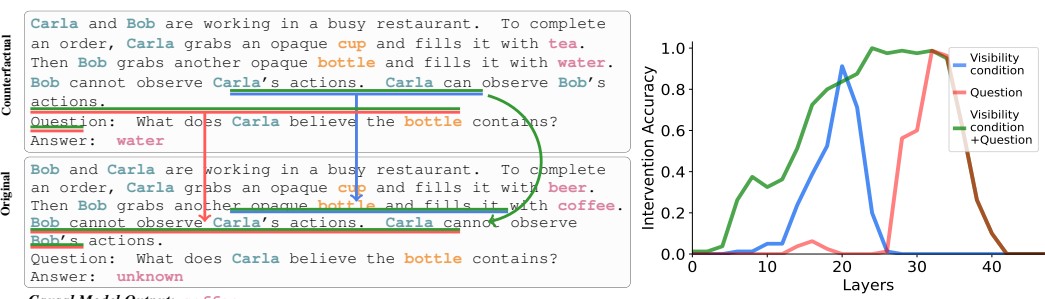

Figure 42: **Visibility Lookback in `Qwen2.5-14B-Instruct`.**

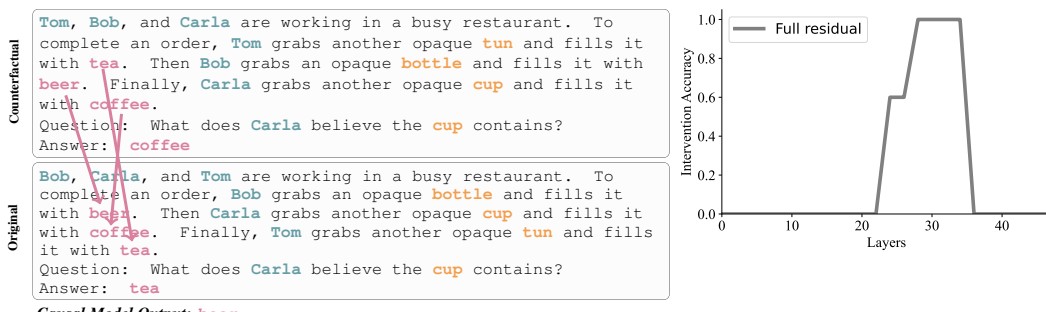

Figure 43: **Payload and address of Binding lookback with three character-object-state triples in `Qwen2.5-14B-Instruct`.**

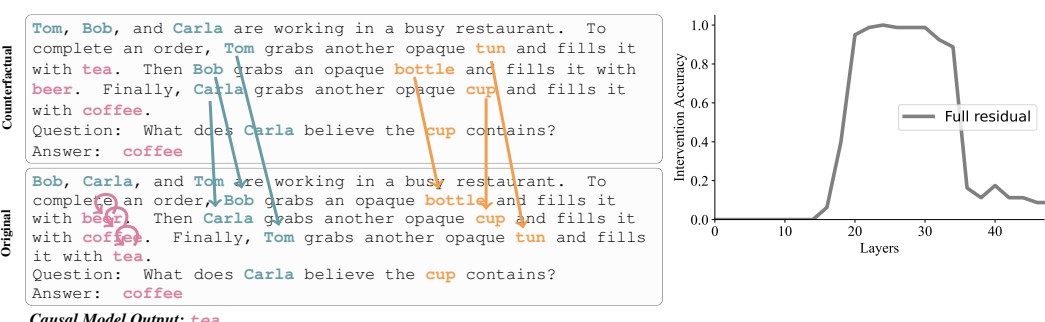

Figure 44: **Source Information of Binding lookback in three character-object-state triples in `Qwen2.5-14B-Instruct`.**

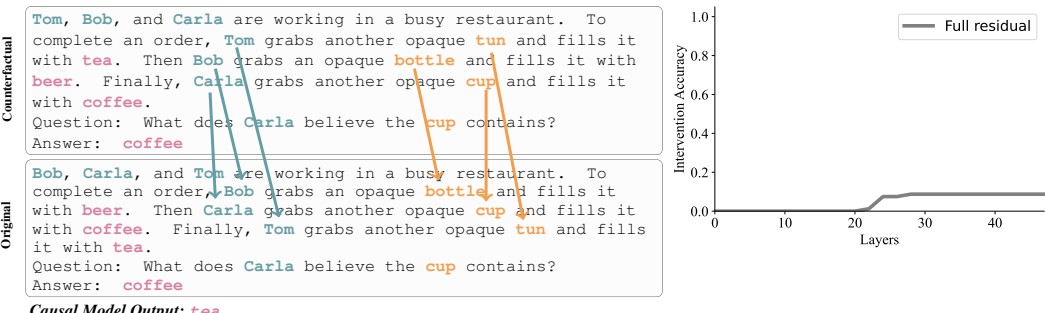

Figure 45: **Source Information of Binding lookback without freezing address and payload in three character-object-state triples in `Qwen2.5-14B-Instruct`.**

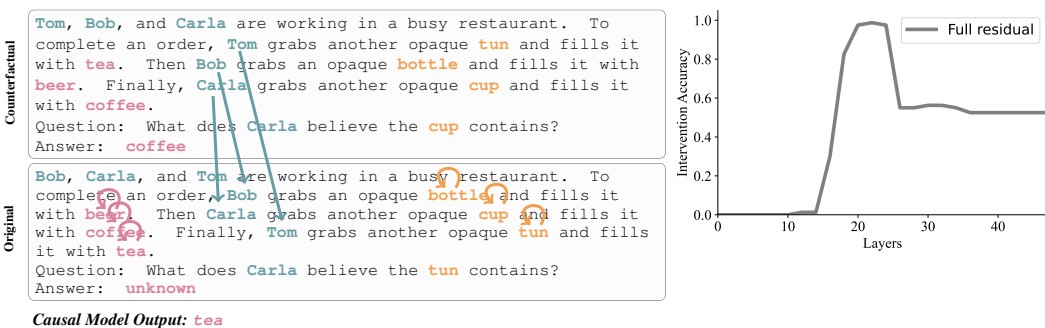

Figure 46: **Character OI in three character-object-state triples in `Qwen2.5-14B-Instruct`.**

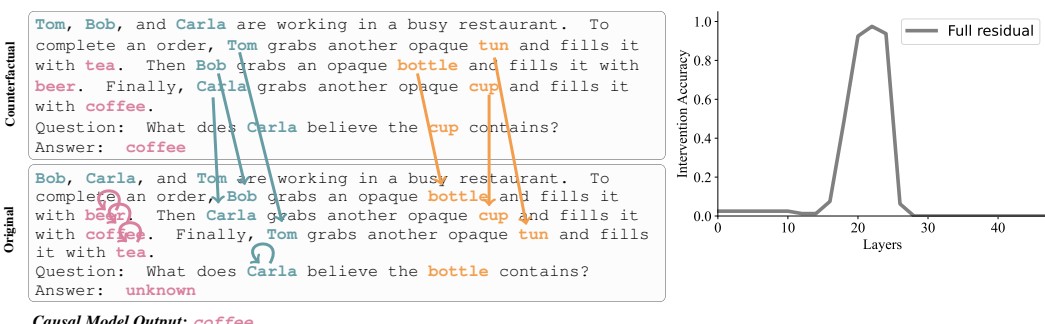

Figure 47: **Object OI in three character-object-state triples in `Qwen2.5-14B-Instruct`.**

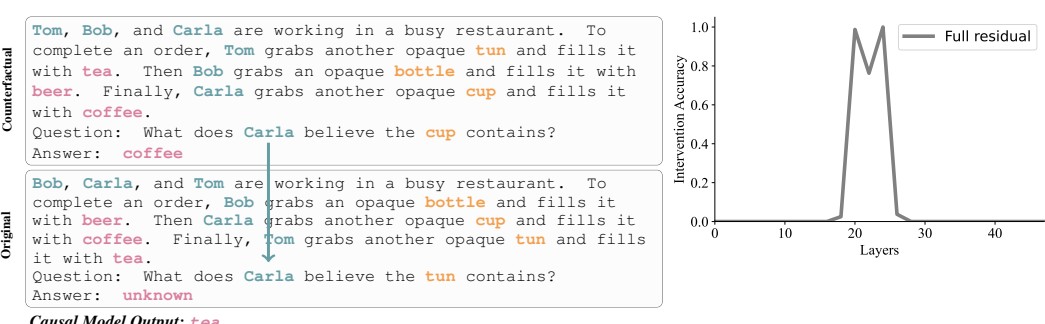

Figure 48: **Query Character OI in three character-object-state triples in `Qwen2.5-14B-Instruct`.**

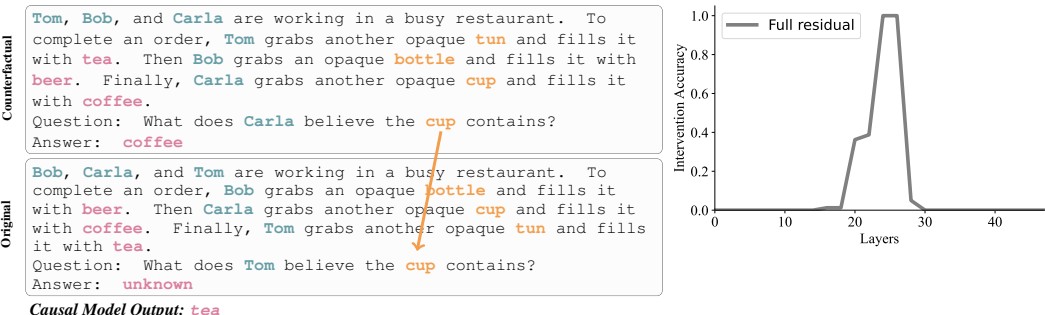

Figure 49: **Query Object OI in three character-object-state triples in `Qwen2.5-14B-Instruct`.**

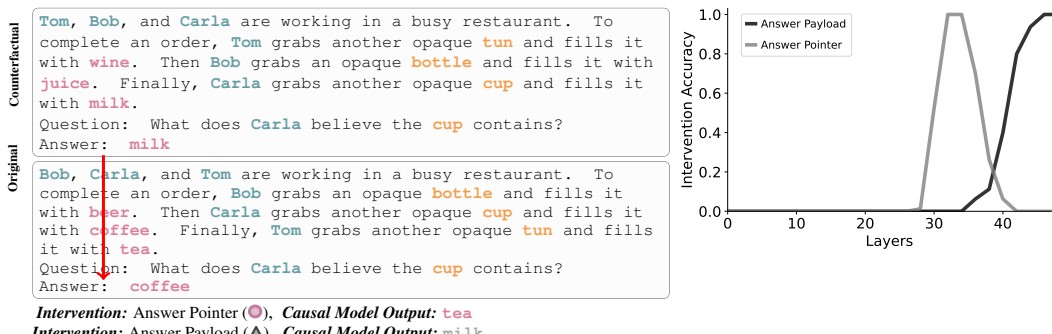

Figure 50: **Answer Lookback Pointer and Payload in three character-object-state triples in `Qwen2.5-14B-Instruct`.**

