# OpenReview forum: "Language Models Use Lookbacks to Track Beliefs"
_ICLR.cc/2026/Conference — ICLR 2026 Poster_

### Official Review · Reviewer_Jrau · 2025-10-24

**Soundness:** 1
**Presentation:** 1
**Contribution:** 1
**Rating:** 0
**Confidence:** 1

**Summary:**

n/a - see ethics review

**Strengths:**

n/a - see ethics review

**Weaknesses:**

n/a - see ethics review

**Questions:**

n/a - see ethics review

**Details Of Ethics Concerns:**

I no longer think it is ethical to work on Theory of Mind in academia, and I think we should be actively discouraging development of these methods. I know that's a strong statement; please bear with me for a minute. I wrote one of the papers referenced heavily works like this, and I have since found out that my work has since been used for unethical applications including persuasion, marketing, and political advertising, in academia and in industry. Yes there are upsides to Theory of Mind – that's why I and many others started working on it in the first place. Better affective technologies, agents that can anticipate your needs without you asking, therapeutic agents, education agents...etc. But I think while these are indeed upsides, we can achieve these goals in other ways that leave the user's agency and mind uninvaded by AI's – e.g., by creating tools that users can have an easier time adapting to and learning, rather than tools that anticipate and adapt to the user. Essentially, I think the upsides are limited and clever solutions could get to the same ends in less invasive ways. And the downsides are basically infinite. If we're successful in building agents that really can understand what we're thinking and feeling without saying it, those agents will be used by big companies and people in power to successfully simulate our thinking and manipulate us into doing what they want. This isn't conjecture or conspiracy theory; each major tech company is actively investigating AI persuasion and marketing for exactly this reason, as are political campaigns and governments throughout the world. I would know, because many of them have reached out to me to help them implement their methods. As people, we should be scared by this.

There's a counter argument of course, which is that this is a cat and mouse game like security, and that academia shouldn't attempt to restrict research into these methods: it's best to have this all out in the open so that people can research defenses. While it's possible that defenses could be created against this type of technology, I think that is a dangerous and unproven hypothesis to rely on. It may be the case that this is a setting in which offense is much easier than defense, especially because unlike in computer security, the defender is ultimately a human, not an algorithm. The best and most secure defense would be to train humans to think critically about these types of attacks, and that is a very hard thing to do, especially as the attacks get more and more sophisticated and human education systems become increasingly politicized and unequal. I think it is much better to keep this pandora's box closed for as long as possible, and to work on functional uses of AI that have fewer ethical downsides.

I would encourage the ethics committee to think carefully about this, discuss among themselves (and maybe reach out to me if you want to chat more), and do as much as possible to prevent works like this from being published in top conferences (which increases the visibility and incentive to work on this subject).  I would also encourage the authors to think carefully about whether they want to contribute to this academic legacy. I worked on this area because I believed in the upsides. I have since regretted my participation in this field and hope it does need lead to bad outcomes; if some day superhuman ToM capabilities come to pass and our society is worse off for it, it will be a burden on all that worked on building it.

---

> ### Author Response · Authors · 2025-11-18
>
> We understand and share the concerns of the reviewer, and in fact, this concern motivates our research. We do not wish to accelerate the capabilities of models to reason about ToM, but rather we wish to understand the mechanisms to enable the detection of ToM behavior by directly reading the neural fingerprint of this type of reasoning. Note that our research intentionally does not focus on amplification of capabilities, but merely measurement and characterization of ToM capabilities that are already present.
>
>
> Our view is that this type of internal neural analysis work is essential if we are concerned that models may be learning to deceive or persuade us.  Because deception and persuasion (by definition) are difficult or impossible to measure from the outputs of a model, we must be able to understand their neural fingerprints so we can detect the difference between superficial outputs that claim some kind of behavior or reasoning, and internal reasoning pathways that directly reveal ToM reasoning.
>
>
> We agree that ToM-like reasoning can be misused (e.g., for persuasion or targeted advertising). However, preventing interpretability work does not mitigate that risk; it worsens it by ensuring that these mechanisms remain hidden and unexamined. As many safety and policy frameworks (e.g., AI auditing standards) emphasize, transparency and mechanistic understanding are key defenses against dual-use risks.
>
>
> We take the ethical implications seriously and have included a section in the revised paper addressing potential misuse. This section clarifies the boundaries of our contribution and emphasizes interpretability’s role as a mitigation, not an enabler, of unethical applications.
>
>
> In summary, our work advances interpretability and safety by providing a transparent account of how existing models track beliefs. We strongly believe restricting such inquiry would hinder oversight and make ToM-like reasoning harder, not easier, to regulate. We hope the committee will view this research as a contribution to responsible and transparent AI science.

---

> > ### Author Response · Authors · 2025-11-25
> >
> > We would like to thank the reviewer again for their feedback. We wanted to check whether our response addresses your concerns. We're happy to provide further clarification if needed.

---

### Official Review · Reviewer_zFrv · 2025-10-30

**Soundness:** 3
**Presentation:** 2
**Contribution:** 4
**Rating:** 6
**Confidence:** 4

**Summary:**

The paper claims evidence for a specific mechanism, the "lookback", that allows a transformer architecture to dereference information about an entity in a sequence of tokens that presents a Theory of Mind problem, so as to correctly answer queries about what the entity knows. The paper's central claim is that the encoding of such information in the residual stream is based on the relative ordering of tokens corresponding to entities in a set of relational statements (first character vs. second character, first object vs. second object), not on structures that encode information as attributes of the entity's identity. The authors present a benchmark dataset (CausalToM) to analyze a model's ability to simulate ToM, and show through causal interventions in the layer-by-layer evolution of the residual stream that ordinal position (encoded as "Ordering IDs"), as opposed to identity, is how the LM manages information to correctly answer ToM queries.

**Strengths:**

The paper addresses an extremely challenging and important problem in understanding the internal mechanisms by which language models perform Theory of Mind reasoning.

The methodology used to address this question is very clearly laid out, and to this reviewer's mind well motivated. The paper provides clear and useful graphical presentations of the mechanism proposed and the results of the interchange interventions in both the no-visibility and visibility cases. The contribution of a structured dataset of simple stories to provide a way to effectively elicit interpretable responses to confirm or disconfirm the Ordering IDs hypothesis is also an important contribution.

The findings demonstrate consistency across models in a given model family (Meta Llama) at multiple sizes (70B and 405B), and preliminary evidence suggests the mechanism generalizes to more naturalistic scenarios (i.e., the BigToM benchmark), strengthening confidence in the robustness of the identified patterns.

**Weaknesses:**

The paper would be significantly strengthened by addressing the following issues related to soundness and presentation. This reviewer sincerely hopes that these can be satisfactorily addressed in the rebuttal phase.

1) The paper's central claim that models use Ordering IDs rather than identity-based or semantic representations is not adequately distinguished from plausible alternatives. There is only the briefest mention of prior work, and it assumes a great deal of familiarity from the reader with what appears to be a very specific body of work. The paper does not explain how the Ordering ID hypothesis was developed, what alternatives were considered, or whether there was exploratory analysis not reported. This makes it very difficult to assess the proposed mechanism in the context of work in mechanistic interpretability to date.

2) The paper lacks crucial information about computational requirements, experimental iteration (how many analyses were attempted before arriving at reported results), and robustness checks (sensitivity to hyperparameters, sample selection, random seeds). Code availability is not mentioned.

3) The paper uses specialized mechanistic interpretability terminology ("residual stream," "QK-circuit," "OV-circuit") without adequate definition, assuming familiarity that may not be universal even among the ICLR community. While core concepts like interchange interventions are explained well, a brief background section defining key architectural concepts would significantly improve accessibility and aid reproducibility.

**Questions:**

Does reducing ToM to positional bookkeeping (lookup mechanism + Ordering IDs) suggest sophisticated behavioral mimicry (à la Searle's Chinese Room thought experiment) rather than understanding or intentionality? What additional evidence would demonstrate conceptual understanding beyond structural pattern extraction? Are the Ordering IDs grounded in meaningful semantic relations about information access and belief formation, or are they arbitrary indices that correlate with correct answers in this constrained task structure?

---

> ### Author Response · Authors · 2025-11-18
>
> **The paper's central claim that models use Ordering IDs...**
>
> We agree with the reviewer’s point that the related work section could benefit from a more thorough discussion. In response, we have expanded this section substantially, situating the concept of Ordering IDs within the broader mechanistic interpretability literature. Our hypothesis, that Ordering IDs appear at character, object, and state tokens, builds on several prior studies suggesting that token-value independent identifiers often emerge that encode their positional structure within the context [1,2,3,4].
>
>
> We provide evidence for the presence of Ordering IDs through a series of experiments. Our work shows that intervening on tokens with identical token values but different Ordering IDs reliably alters the model’s internal computation, producing systematic shifts in the final outputs predicted by our high-level causal model. This indicates that the LM uses Ordering IDs as structural building blocks, incorporating them into lookback mechanisms that track, update, and retrieve beliefs throughout the sequence.
>
> ---
>
> **The paper lacks crucial information about computational requirements...**
>
> We agree with the reviewer that additional implementation details should be incorporated into the manuscript. Hence, we have included the Reproducibility Statement that contains the details about computational requirements and an anonymous link to the code repository, which contains all hyperparameters, random seeds, and dataset generation details for reproducibility of the work.
>
>
> Our research followed a highly iterative experimental process, evolving from broad benchmarking to fine-grained, hypothesis-driven causal abstraction analysis. We began by evaluating LLMs on standard ToM datasets. This initial analysis revealed that open-weight models aren’t very good at ToM benchmarks. Additionally, most of the benchmarks were unsuitable for causal analysis. Hence, we decided to create the synthetic dataset CausalToM, which larger llama models were able to solve coherently while incorporating ToM characteristics.
>
>
> Once a stable task and model were identified, we iteratively applied a sequence of mechanistic interpretability techniques, starting with direct logit attribution, attention pattern visualization, and attention knockout to understand the computation at the final token. During this phase, we discovered that LMs use positional information, similar to multiple existing works in the literature, to retrieve the correct state token. This was followed by more granular analyses with the Causal Abstraction framework and residual vector interchange intervention. These intervention experiments were themselves iterated upon; for example, we tested different token positions (e.g., punctuation, preceding tokens) to find where information was stored. Each finding led to a more refined hypothesis, which was then tested with new, targeted patching experiments to identify specific computational variables. We had to follow this deeply iterative hypothesis-driven approach to uncover the end-to-end mechanism of belief tracking in LMs.
>
> ---
>
> **The paper uses specialized mechanistic interpretability terminology ...**
>
> We thank the reviewer for this insightful feedback. In response, we have added a new section to the Appendix (Appendix C) that describes the residual stream framework of the transformer architecture, including the QK- and OV-circuit, which is referenced from Section 2. It should aid the readers of varied backgrounds in comprehending the method and results.
>
> ---
>
> [1] How do Language Models Bind Entities in Context?, ICLR 2024.
>
> [2] Fine-Tuning Enhances Existing Mechanisms: A Case Study on Entity Tracking, ICLR 2024.
>
> [3] Emergent Symbolic Mechanisms Support Abstract Reasoning in Large Language Models, ICML 2025.
>
> [4] Representational Analysis of Binding in Language Models, ACL 2024.

---

> > ### Author Response · Authors · 2025-11-18
> >
> > **Does reducing ToM to positional bookkeeping (lookup mechanism + Ordering IDs) suggest sophisticated behavioral mimicry (à la Searle's Chinese Room thought experiment) rather than understanding or intentionality?**
> >
> > We thank the reviewer for raising this philosophical question. Our position is that the mechanisms we uncover point toward algorithmic computation within the model, not toward a Chinese-Room–style lookup procedure.
> >
> > Searle’s argument hinges on a system that merely shuffles symbols via an enormous lookup table, with no intermediate reasoning or rule-governed transformations. By contrast, the structures we observe in the model involve coordinated internal procedures: the model constructs intermediate variables, assigns them functional roles (e.g., pointer vs. address), and uses them in multi-step dereference-like computations that unfold across layers. These are not static associations but dynamically executed operations, with intermediate states that causally mediate later ones. This is the hallmark of an algorithmic process, not a lookup table.
> >
> > In other words, even if the model lacks intentionality in a philosophical sense, its internal organization is not that of Searle’s Room. It is not merely simulating understanding by memorizing all possible pairings of situations and answers; instead, it is executing a series of learned operations that extract, bind, and retrieve abstract variables.
> >
> > ---
> >
> > **What additional evidence would demonstrate conceptual understanding beyond structural pattern extraction?**
> >
> > Our paper is an initial step, and there are several further questions that remain to be clarified to demonstrate further conceptual understanding of ToM. One is how the model reasons about false beliefs, with more than two agents, and a range of other settings, including the special subjects "you" and "me"; if future experiments show that the same algorithms are employed across these different settings, it will provide stronger evidence of a conceptual understanding. The other key will be to study the interaction with goal-setting and goal-seeking mechanisms, such as the ability to reason about creating a belief in another agent.
> >
> > ---
> >
> > **Are the Ordering IDs grounded in meaningful semantic relations about information access and belief formation, or are they arbitrary indices that correlate with correct answers in this constrained task structure?**
> >
> > We appreciate the reviewer’s question. Our findings indicate that Ordering IDs (OIs) are not arbitrary indices but semantically structured internal variables that the model uses to represent distinct entity types relevant to belief formation. Character OIs, object OIs, and state OIs occupy separate low-rank subspaces, and they cannot be interchanged under intervention without disrupting the model’s reasoning. This separation suggests that OIs encode meaningful semantic type distinctions rather than arbitrary positional tags.
> >
> > Moreover, these type-specific OIs support systematic generalization across counterfactual stories, which implies that they serve as functional building blocks in the model’s internal ontology. A promising direction for future work is assessing whether similar OIs emerge in naturalistic text settings, where entity types and belief dependencies are less explicitly scaffolded. Such an investigation could help determine how broadly these semantically grounded OI subspaces operate in real-world contexts.

---

> > > ### Author Response · Authors · 2025-11-25
> > >
> > > We would like to thank the reviewer again for their feedback. We wanted to check whether our response addresses your concerns. We're happy to provide further clarification if needed.

---

> > > ### Comment · Reviewer_zFrv · 2025-11-26
> > > **Responses to my questions with respect to intentionality and semantics are addressed**
> > >
> > > I want to thank the authors for their thoughtful response to my philosophical queries. They are consistent with my impression, in reading the literature on LLMs, meaning, and intentionality that has been emerging over the last few years, that LLMs exist in something like a halfway point between being "stochastic parrots" and having intentionality in the sense that humans do. The argument from there being a process that exhibits variable-binding-like behavior is plausible. I also appreciate the fact that the authors acknowledge that there is much clarification to be accomplished before we can truly speak definitively about such behavior. This is in keeping with my current contribution scoring.

---

> > ### Comment · Reviewer_zFrv · 2025-11-26
> > **Concerns with respect to comments in the weakness section have been addressed**
> >
> > I appreciate the authors' addressing the points I raised around adding content in the paper to make it easier for those unfamiliar with the research context and terminology to better understand the results, and adding details in support of reproducibility. These changes address my concerns as raised in my review, and I am adjusting my presentation score upwards.

---

### Official Review · Reviewer_AD4K · 2025-10-31

**Soundness:** 3
**Presentation:** 4
**Contribution:** 2
**Rating:** 6
**Confidence:** 4

**Summary:**

This paper investigates the internal mechanisms by which LLMs track characters' beliefs in ToM tasks.  The authors construct CausalToM, employ causal mediation analysis and causal abstraction to identify systematic computational patterns. Three specific lookback mechanisms are identified: (1) binding lookback that links character-object-state triples via ordering IDs, (2) answer lookback that retrieves state token values, and (3) visibility lookback that updates beliefs based on character observability. The mechanisms are validated through interchange intervention experiments on Llama-3-70B-Instruct and Llama-3.1-405B-Instruct models.

**Strengths:**

Unlike previous works in the Theory of Mind (ToM) domain, such as prompt-based (Think twice, TimeToM), tool-based (Social world model), or model-based approaches (Bayesian framework), this paper analyzes the model’s belief reasoning ability from a novel and interpretable perspective. In ToM research, there has long been debate over whether models’ ToM abilities are truly robust, and whether a correct answer to a ToM question genuinely reflects capabiltiy level. Analyzing this issue from the viewpoint of interpretability offers a promising path toward resolving this controversy. The paper presents excellent visualizations and provides a clear description of the research background.

**Weaknesses:**

The data pattern of CausalToM mentioned in the paper is quite simple.

Theory of Mind (ToM) is a broad framework encompassing various dimensions of mental states, and its scenarios are often diverse and complex. The interpretability analysis in this paper is applied only to a narrow data scope (simple story settings and the belief dimension). When the data scenarios become more complex (e.g., longer narratives or richer social contexts), can this method still maintain good scalability and generalization?

Moreover, the interpretability analysis is conducted only on the LLaMA series models. Will these observed phenomena also appear in other model families?

**Questions:**

See Weakness.

The models used by the authors are 70B and 405B. Would the same phenomena described in the paper also appear if the model size were around 7–32B?

---

> ### Author Response · Authors · 2025-11-18
>
> **When the data scenarios become more complex (e.g., longer narratives or richer social contexts), can this method still maintain good scalability and generalization?**
>
> We thank the reviewer for raising this good point. Challenges certainly remain, but we believe that formulating high-level causal hypotheses and testing them through interchange intervention experiments is a sound approach to understanding complex systems such as LMs. For example, our approach of first examining a synthetic environment like CausalToM and then applying the resulting insights to a more complex setting like BigToM [1] helps lay the groundwork for disentangling the intricate internal processes of LMs, which is more scalable and generalizable.
>
> That said, the path ahead is far from straightforward. As context lengths grow, interchange intervention experiments become increasingly difficult because information is distributed across many tokens. Crucially, this difficulty reflects a broader limitation of the mechanistic interpretability field rather than a constraint specific to our work [2, 3]. We strongly support ongoing efforts to broaden existing mechanistic interpretability techniques, most of which have been validated only on relatively narrow data distributions, so they can be applied effectively in more realistic, real-world scenarios.
>
> [1] - Understanding Social Reasoning in Language Models with Language Models, NeurIPS 2023.
>
> [2] - Challenges in Mechanistically Interpreting Model Representations, ICML 2024 Workshop Mechanistic Interpretability Workshop.
>
> [3] - Open Problems in Mechanistic Interpretability, TMLR 2025.
>
> ---
>
> **The models used by the authors are 70B and 405B. Would the same phenomena described in the paper also appear if the model size were around 7–32B? Generalization across model families?**
>
> We are happy to report that several newer models, particularly the recently released Qwen series, are now able to solve the CausalToM task consistently, complementing the strong performance previously observed only in the larger Llama models. In the early stages of this project, we found that coherent behavior on the CausalToM task (above 80%) was limited to Llama models with sufficiently large parameter counts. Following your suggestion, we recently reassessed the behavioral performance of a wide range of newer models across multiple families, over 100 samples (10 runs). The table below summarizes these results. In addition to Llama models above 70B parameters, the latest Qwen models also demonstrate coherent and stable performance on the task, marking an encouraging expansion in the set of models capable of handling this type of reasoning.
>
>
> Despite this progress, most other models across several families, including Llama, Qwen, Olmo, and Gemma, still struggle to perform the task reliably. For this reason, we conducted the interchange intervention experiments described in the paper on Qwen2.5-14B-Instruct, one of the strongest mid-sized models identified in our re-evaluation. Results are reported in Appendix N.
>
>
> The experimental outcomes aligned with our expectations: the mechanism uncovered in the original study generalizes cleanly to this new model, suggesting that it captures a robust and potentially universal structure underlying how LMs perform belief tracking. These findings strengthen the broader claim that the mechanism is not tied to a single architecture or scale but instead reflects a deeper computational pattern.
>
>
> We have incorporated these new results into the appendix and updated the main text to reflect the expanded set of behavioral and mechanistic evaluations.
>
> | Model Name | No Visibility (mean ± std) | Visibility (mean ± std) |
> |------------|---------------------------|-------------------------|
> | **7B Models** | | |
> | Llama-2-7b-hf | 0.011 ± 0.007 | 0.006 ± 0.008 |
> | Qwen2.5-7B | 0.280 ± 0.039 | 0.061 ± 0.017 |
> | Qwen2.5-7B-Instruct | 0.948 ± 0.022 | 0.719 ± 0.031 |
> | **8B Models** | | |
> | Llama-3.1-8B | 0.446 ± 0.046 | 0.293 ± 0.042 |
> | Llama-3.1-8B-Instruct | 0.722 ± 0.044 | 0.310 ± 0.035 |
> | Meta-Llama-3-8B | 0.297 ± 0.027 | 0.117 ± 0.024 |
> | Meta-Llama-3-8B-Instruct | 0.349 ± 0.042 | 0.085 ± 0.025 |
> | **13B Models** | | |
> | Llama-2-13b-hf | 0.328 ± 0.041 | 0.117 ± 0.027 |
> | OLMo-2-1124-13B-Instruct | 0.522 ± 0.034 | 0.36 ± 0.049 |
> | **14B Models** | | |
> | Qwen2.5-14B | 0.865 ± 0.038 | 0.433 ± 0.040 |
> | **Qwen2.5-14B-Instruct** | **0.962 ± 0.021** | **0.912 ± 0.022** |
> | **27B Models** | | |
> | Gemma-3-27b-it | 0.527 ± 0.036 | 0.388 ± 0.034 |
> | **32B Models** | | |
> | OLMo-2-0325-32B-Instruct | 0.814 ± 0.033 | 0.679 ± 0.029|
> | **70B Models** | | |
> | **Meta-Llama-3-70B-Instruct** | **0.952 ± 0.020** | **0.923 ± 0.014** |
> | **405B Models** | | |
> | **Meta-Llama-3.1-405B-Instruct** | **0.883 ± 0.041** | **0.97 ± 0.013** |

---

> > ### Author Response · Authors · 2025-11-25
> >
> > We would like to thank the reviewer again for their feedback. We wanted to check whether our response addresses your concerns. We're happy to provide further clarification if needed.

---

> > > ### Comment · Reviewer_AD4K · 2025-11-27
> > > **Response to Rebuttal**
> > >
> > > Thank you to the authors for the supplementary experiments. I believe that better understanding the interpretability of LLMs in the ToM domain is important and meaningful (beyond only QA evaluation), and it will contribute to the healthier development of this field. Moreover, the ToM capability of LLMs, or social intelligence, is a crucial dimension in the foreseeable future.
> > >
> > > I will increase my score to 8.

---

### Official Review · Reviewer_dpdh · 2025-11-03

**Soundness:** 3
**Presentation:** 3
**Contribution:** 3
**Rating:** 6
**Confidence:** 3

**Summary:**

This paper asks a concrete mechanistic question pertaining to ToM: how transformers store, update, and retrieve characters and their states. The dataset is CausalToM, a toy two‑sentence story set with two characters, two objects (containers), and two object states (contents). Each example concludes with a question, such as “What does Bob believe the bottle contains?”, accompanied by optional visibility statements that specify who can observe whom. The authors analyze Llama‑3‑70B‑Instruct using interchange interventions on residual activations (i.e., patching counterfactual activations) to observe how the model’s behavior changes.

The central finding is a pervasive lookback mechanism. The model “writes” tags (my terminology; referred to as OIs in the paper) for the “first/second” character, object, and state into the residual stream. Further, state tags are bound to the appropriate character/object tags at the state token, and the model then “looks back” from the answer position to retrieve (i) the right state tag and then (ii) the actual state token via attention. The lookback is localized layer-wise (e.g., tags form approximately layers 20–34; binding occurs at the state token, approximately layers 33–38). Figures 1, 3, 4, and 7 illustrate the pointer/address/payload flow and the three lookbacks.

**Strengths:**

Overall, the results are surprising and insightful. The analysis is not hand-wavy.

* Strong causal methodology. The authors use interchange interventions (activation patching) with carefully matched counterfactual stories to manipulate specific internal variables and measure IIA layer by layer. For example, patching the final “Answer:” token at mid layers redirects the answer pointer (layers 34–52), whereas patching late layers swaps the answer payload (at layers> 56).



* Careful dataset design for causal analysis. CausalToM is deliberately simple (two characters/objects/states), so counterfactuals differ in only one factor at a time.

**Weaknesses:**

1. Analysis restricted to successful cases. All mechanistic experiments are run on 80 correctly answered examples. This risks selection bias: we only study the circuit when it worked. What happens for incorrect cases?


2. Scaling beyond “first/second” is unclear. tags/OIs encode first vs. second character/object/state, which is perfect for this dataset, but what happens as you scale up?

A small toy study (e.g., 3+ entities per type) would help address both weaknesses by revealing failure modes and testing whether the mechanism extends beyond binary order.

**Questions:**

Please see weaknesses.

---

> ### Author Response · Authors · 2025-11-18
>
> **Analysis restricted to successful cases…**
>
> The reviewer raises a valuable point about our decision to filter out examples where the model fails to answer correctly. Our goal was to isolate the mechanism responsible for successful belief tracking, and filtering out unsuccessful runs is a standard approach in mechanistic interpretability research [1, 2, 3]. We agree that investigating failure modes (e.g., understanding when and how the mechanism breaks down) is an important and natural next step, and leave it for future work. Please note that, in order to analyse failures of the mechanism, one should start by revealing the mechanism responsible for successful reasoning. In this sense, our work laid the groundwork for such follow-up studies.
>
> ---
>
> **Scaling beyond “first/second” is unclear…**
>
> We agree with the reviewer that examining the belief-tracking capabilities of language models in scenarios involving three or more entities would be highly informative. Accordingly, we evaluated the best-performing models on three-entity settings, i.e., three characters, three objects, and three states. The table below reports their behavioral performance in both the no-visibility and visibility conditions, evaluated over 100 samples (10 runs). Because these models can only solve the no-visibility setting coherently, we focused our analysis of their internal mechanisms in this setting. Specifically, we conducted interchange intervention experiments on Qwen2.5-14B-Instruct. As reported in Appendix O, the results show that it uses the same mechanism, binding, and answer lookback previously identified.
>
> | Model Name | No Visibility (mean ± std) | Visibility (mean ± std) |
> |------------|---------------------------|-------------------------|
> | **Qwen2.5-14B-Instruct** | 0.945 ± 0.025 | 0.243 ± 0.036 |
> | **Meta-Llama-3-70B-Instruct** | 0.878 ± 0.027 | 0.247 ± 0.035 |
> | **Meta-Llama-3.1-405B-Instruct** | 0.864 ± 0.051 | 0.233 ± 0.05 |
>
> For the camera-ready version, we will also report experiments on both Llama models on the three-entity settings and include the results in the appendix.
>
>
> [1] - Mechanistic Interpretability of Emotion Inference in Large Language Models, ACL 2025.
>
> [2] - MIB: A Mechanistic Interpretability Benchmark, ICML 2025.
>
> [3] - Interpretability Analysis of Arithmetic In-Context Learning in Large Language Models, EMNLP 2025.

---

> > ### Author Response · Authors · 2025-11-25
> >
> > We would like to thank the reviewer again for their feedback. We wanted to check whether our response addresses your concerns. We're happy to provide further clarification if needed.

---

### Author Response · Authors · 2025-11-18

We thank all reviewers for their thoughtful and constructive feedback. We have incorporated your suggestions into the revised manuscript and uploaded an updated version. Your recommendations have significantly strengthened the work. New results (described below) show that the **discovered belief-tracking mechanism generalizes across model families and scales, as well as to more complex settings involving three character–object–state triples**.

We summarize our clarifications and updates below:

 - **Updated evaluation of newer LLMs on CausalToM in Appendix D:**
We expanded our behavioral evaluation to include multiple recently released models across several families, including Llama, Qwen, Olmo, and Gemma. We found that, beyond the Llama models reported in the original submission, only the newer Qwen models (particularly Qwen2.5-7B-Instruct and Qwen2.5-14B-Instruct) can perform CausalToM consistently in both visibility and no-visibility settings.

 - **Mechanism generalizes to additional model families and scales (Appendix N):**
We conducted interchange intervention experiments on Qwen2.5-14B-Instruct. The same lookback-based mechanism emerged cleanly, reinforcing that the identified computation is not tied to a single architecture or scale.

 - **Mechanism generalizes to a more complex setting with three character-object-state triplets (Appendix O).**
We also conducted interchange intervention experiments in the no-visibility setting using three character–object–state triples. The results indicate that the LM relies on the same mechanism, combining binding and answer lookback, to solve the task.

 - **Reproducibility and implementation details added:**
We have added a Reproducibility Statement describing computational requirements and dataset creation details. We have also provided a link to an anonymous code repository containing the full implementation and all experimental configurations.

 - **Ethics Statement included:**
We have included a detailed Ethics Statement, discussing the dual-use concerns related to ToM-like reasoning and explaining how interpretability-focused analyses help promote transparency, auditing, and safety rather than capability enhancement.


We appreciate the reviewers’ insights and believe that these revisions meaningfully strengthen the clarity, transparency, and soundness of the work.

---

### Author Response · Authors · 2025-12-03
**Authors’ Final Remarks**

We sincerely thank the reviewers for their detailed and constructive feedback. We are encouraged by their positive reception of our work, including praise for its experimental design (“Strong causal methodology”, “Careful dataset design for causal analysis”), its contribution to the field (“Analyzing this issue from the viewpoint of interpretability offers a promising path toward resolving this controversy”), and its presentation (“The paper presents excellent visualizations and provides a clear description of the research background”).

In addition to these positive evaluations, the reviewers raised some concerns, which we addressed in our responses and by expanding the revised manuscript in the following ways:
* Expanded experimental validation: We added both behavioral and mechanistic evidence demonstrating that our belief-tracking mechanism generalizes across model families, scales, and multi-entity settings.
* Improved clarity: We expanded the related-work section and added brief background explanations of key interpretability concepts.
* Enhanced reproducibility: We included a Reproducibility Statement, which provides a link to an anonymous code repository and other implementation details.
* Stronger ethical framing: We also added an explicit Ethics Statement emphasizing interpretability’s role in auditing and safety rather than capability amplification.

We appreciate the reviewers’ thoughtful feedback and believe the resulting revisions significantly strengthen the clarity and generalizability of our contributions. We are grateful for the constructive dialogue and the improvements it enabled, and we thank the reviewers for the time and expertise they invested in guiding this work.

Sincerely,

Authors

---

### Meta-Review · Area_Chair_SZs5 · 2026-01-08

**Summary:**

The paper presents a thorough mechanistic interpretability study of how language models track character beliefs in Theory of Mind tasks. Reviewers appreciated the strong causal methodology using interchange interventions, careful dataset design (CausalToM), and excellent visualizations. Key concerns included limited scaling analysis (only two entities) and restriction to successful cases. Authors addressed these through extensive new experiments showing the identified lookback mechanism generalizes across Llama and Qwen model families at multiple scales, and extends to three-entity settings.

**Reviewer Concerns:**

All major reviewer concerns were effectively addressed in the rebuttal. Reviewer dpdh's concern about scaling beyond two entities was resolved by demonstrating that the binding and lookback mechanisms work identically in three-entity scenarios. Reviewer AD4K's concern about generalization across model scales was addressed by showing the same mechanisms emerge in Qwen2.5 models at 7B-14B parameter scales, not just in large Llama models. Reviewer zFrv's concerns about presentation clarity and reproducibility were resolved by adding terminology definitions, background sections, and a reproducibility statement with code availability. One outstanding minor point: investigating failure modes and performance across even more diverse model architectures remains future work.

**Reviewer Scores:**

Reviewer dpdh (initial rating 6): The supplementary experiments on three-entity settings would likely increase the score to 7-8, as the scaling concern was explicitly addressed.
Reviewer AD4K (initial rating 6, updated to 8): Already increased their score after rebuttal; the mechanism generalization across model families and scales fully addressed their concerns.
Reviewer zFrv (initial rating 6): The additions to paper clarity and reproducibility would likely increase the score to 7-8, as these concerns were directly resolved. Overall, the rebuttal substantially strengthened the work, and all three reviewers have acknowledged that the authors' responses meaningfully addressed their concerns.

---

### Decision · Program_Chairs · 2026-01-26

Accept (Poster)